# SpanNorm: Reconciling Training Stability and Performance in Deep Transformers

Chao Wang [* 1]  Bei Li [* 1]  Jiaqi Zhang [1]  Xinyu Liu [2]  Yuchun Fan [2]  Linkun Lyu [1]  Xin Chen [1]  Jingang Wang [1]
Tong Xiao [2]  Peng Pei [1]  Xunliang Cai [1]

## Abstract

The success of Large Language Models (LLMs) hinges on the stable training of deep Transformer architectures. A critical design choice is the placement of normalization layers, leading to a fundamental trade-off: the "PreNorm" architecture ensures training stability at the cost of potential performance degradation in deep models, while the "PostNorm" architecture offers strong performance but suffers from severe training instability. In this work, we propose SpanNorm, a novel technique designed to resolve this dilemma by integrating the strengths of both paradigms. Structurally, SpanNorm establishes a clean residual connection that spans the entire Transformer block to stabilize signal propagation, while employing a PostNorm-style computation that normalizes the aggregated output to enhance model performance. We provide a theoretical analysis demonstrating that SpanNorm, combined with a principled scaling strategy, maintains bounded signal variance throughout the network, preventing the gradient issues that plague PostNorm models, and alleviating the representation collapse of PreNorm. Empirically, SpanNorm consistently outperforms standard normalization schemes in both dense and Mixture-of-Experts (MoE) scenarios, paving the way for more powerful and stable Transformer architectures.

## 1. Introduction

The Transformer architecture (Vaswani et al., 2017) has become the cornerstone of modern natural language pro-

cessing, demonstrating unprecedented success and scalability, particularly in the realm of Large Language Models (LLMs) (Brown et al., 2020; Touvron et al., 2023). The remarkable capabilities of these models are not merely a function of their attention mechanisms but are also deeply rooted in fundamental design choices that ensure stable and effective training at scale. Among these, the interplay between residual connections (He et al., 2016) and normalization layers (Lei Ba et al., 2016) stands out as a critical factor for success. While both are integral, our work focuses on advancing the understanding and application of normalization within deep Transformer networks.

The evolution of normalization in Transformers provides a compelling narrative of balancing performance and stability. The original Transformer design employed a "PostNorm" architecture, where Layer Normalization (LN) is applied after the residual connection. This configuration is often associated with strong performance in shallower models but becomes increasingly difficult to train as network depth increases, suffering from vanishing and exploding gradients that prevent convergence beyond a few dozen layers (Wang et al., 2019; Xiong et al., 2020). To overcome this limitation, the "PreNorm" architecture was proposed, which moves the normalization layer to the input path of each sub-layer, before the residual addition. This simple yet effective change ensures a clean, unnormalized residual path, which significantly stabilizes gradient flow and enables the training of models with tens of layers (Wang et al., 2019).

Consequently, the PreNorm Transformer has become the de-facto standard for training deep LLMs (Touvron et al., 2023; Liu et al., 2024; Yang et al., 2025). However, this stability can come at a cost, as PreNorm models sometimes exhibit a "depth degradation" problem, where performance does not improve, or even degrades with increased depth (Wang et al., 2024; Li et al., 2020; Liu et al., 2020; Sun et al., 2025). This issue is frequently attributed to the tendency of PreNorm architectures to optimize an ensemble of shallower subnetworks, an effect facilitated by their direct residual paths. While effective to a point, this ensemble approach eventually hits a performance ceiling as network depth increases. Conversely, PostNorm models are believed to learn more

---

[*]Equal contribution  [1]Meituan Inc.  [2]NLP Lab, School of Computer Science and Engineering, Northeastern University, Shenyang, China. Correspondence to: Jiaqi Zhang <zhangjiaqi39@meituan.com>.

*Proceedings of the 43rd International Conference on Machine Learning*, Seoul, South Korea. PMLR 306, 2026. Copyright 2026 by the author(s).

potent, deeply integrated features but are notoriously hampered by training instability. This phenomenon has spurred a new wave of research. On one hand, efforts have been made to stabilize PostNorm training to harness its performance benefits in deeper networks (Shleifer et al., 2021; Liu et al., 2020; Wang et al., 2024; Zhuo et al., 2025). On the other hand, researchers are exploring ways to mitigate the performance limitations of PreNorm architectures (Kim et al., 2025; Sun et al., 2025).

In this work, we introduce SpanNorm, a novel normalization technique that directly addresses this trade-off. As the name implies, this architecture establishes a direct structural *span* from the block input to the final normalization, bypassing the intermediate layers. Concretely, SpanNorm adopts the core design principle of PreNorm by establishing a clean path between layers, while adopting the PostNorm-style computation that normalizes the sum of the layer's input and the residual branch output. A comparative analysis of gradient dynamics reveals the advantages of SpanNorm over both PreNorm and PostNorm.

To further ensure robust training for large-scale models, we also outline principles for an effective scaling strategy, encompassing both depth and width scaling. Our analysis demonstrates that SpanNorm, when paired with appropriate initialization and learning rate schedules, effectively preserves signal norms as information propagates through the network. We specifically show that the gradient signals avoid the vanishing gradient problem. Our experimental results validate this stability across a wide range of configurations, confirming that SpanNorm provides a solid foundation for training Transformers with varying depths and parameter scales, covering both standard dense architectures and MoE models.

Our main contributions are summarized as follows:

- We propose SpanNorm, a novel normalization strategy that harmonizes the training stability of PreNorm with the superior performance of PostNorm. By establishing a clean residual path while normalizing the residual sum, SpanNorm effectively resolves the dilemma between optimization stability and representational capacity.

- We provide a theoretical analysis of gradient dynamics, proving that SpanNorm avoids both the vanishing gradients of PostNorm and the representation collapse of PreNorm. Furthermore, we derive a theoretically justified initialization strategy (Scale Init) that guarantees bounded signal variance, enabling stable training even as depth increases.

- We demonstrate through extensive experiments on both dense and Mixture-of-Experts (MoE) models that SpanNorm consistently outperforms state-of-the-art normalization counterparts. Notably, we successfully scale SpanNorm to an ultra-deep 128-layer model (6.5B), demonstrating robust training stability and superior performance. Crucially, SpanNorm incurs no additional computational overhead, serving as a strictly better, plug-and-play replacement for existing norms in large-scale LLMs.

**Conflict of Interest Disclosure**   The authors declare that they have no competing interests.

## 2. Related Work

The placement of Layer Normalization (LN) is critical to Transformer stability. The original PostNorm design applies normalization after the residual connection, performing effectively in shallower models, but suffering from unstable gradients in deep networks. A simple yet impactful modification, PreNorm, reverses this order by applying normalization to the input of the sub-layer, maintaining a "clean" residual path that mitigates vanishing gradients (Wang et al., 2019) and removes the need for aggressive warm-up schedules (Xiong et al., 2020; Liu et al., 2020). This stability advantage made it the standard in Vision Transformer (ViT) (Dosovitskiy et al., 2021), GPT-3 (Brown et al., 2020), and most modern large language models (Touvron et al., 2023; Yang et al., 2025; Team et al., 2025).

However, deep PreNorm models exhibit their own limitations. The clean residual path can lead to representation collapse, where upper layers fail to learn new features (Li et al., 2020). This "Curse of Depth" is characterized by exponential growth in activation variance and layers that devolve into identity functions, limiting the benefits of scaling depth (Sun et al., 2025).

These challenges spurred two research streams. The first focuses on improving PreNorm. Approaches like Normformer (Shleifer et al., 2021) introduce auxiliary normalization layers to balance gradient norms, while LayerNorm Scaling (Sun et al., 2025) explicitly regulates activation variance by rescaling LN outputs. More recently, "sandwich" designs that wrap blocks with normalization both before and after the attention mechanism have also demonstrated potential (Team et al., 2024; Kim et al., 2025). The second stream aims to stabilize PostNorm to harness its performance. Zhang et al. (2019) showed that a depth-scaled initialization strategy could alleviate the gradient vanishing problem, and Huang et al. (2020) later provided theoretical support, demonstrating that with careful initialization, deep PostNorm models can be optimized effectively. Methods like DeepNorm (Wang et al., 2024) have successfully trained thousand-layer Transformers using specific rescaling and initialization strategies, highlighting PostNorm's sensitivity to these choices.

Despite the aforementioned studies, several hybrid ap-

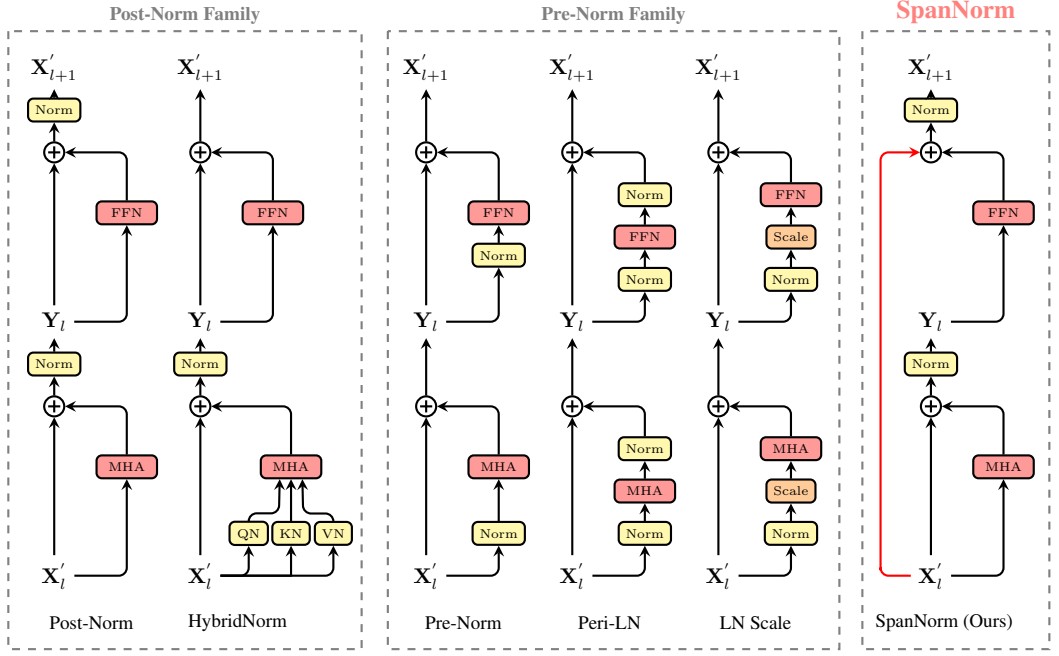

*Figure 1.* Comparisons of our proposed SpanNorm with PostNorm, PreNorm, and other advanced variants. Here, we take the dense model as an instance, and MHA denotes the multi-head attention, FFN denotes the feedforward network. Note that MHA can also be replaced by GQA, MLA and other attention variants. When switching to the MoE models, FFN could be replaced by MoE modules.

proaches have emerged to combine the benefits of both paradigms. Mix-LN (Li et al., 2024) adopts an inter-layer hybrid strategy, using PostNorm for shallower layers and transitioning to PreNorm for deeper layers. In contrast, HybridNorm (Zhuo et al., 2025) utilizes an intra-layer approach, employing a QKV-Norm (where normalization is applied on the head dimension) before the attention computation while retaining a PostNorm after the attention output's residual connection, thereby attempting to gain stability without sacrificing performance.

## 3. SpanNorm: Design and Analysis

In this section, we introduce SpanNorm, a novel modification to the Transformer block architecture designed to enhance training stability and performance. We begin by reviewing the standard PreNorm and PostNorm structures, and then present the SpanNorm architecture and provide a theoretical analysis of its advantages, focusing on its benefit against the PreNorm and PostNorm models. The overall architecture of SpanNorm and other variants is shown in Figure 1.

### 3.1. Preliminaries: Normalization in Transformers

The placement of layer normalization within a Transformer block is a critical design choice that significantly impacts model training dynamics and final performance. The two dominant strategies are PostNorm and PreNorm.

**Post-LayerNorm (PostNorm):** The original Transformer architecture introduced the PostNorm variant. In this setup, LayerNorm is applied after the residual connection in each sub-layer. The computation flow for a single block is:

$$\mathbf{Y}_l = \mathrm{LN}(\mathrm{MHA}(\mathbf{X}'_l) + \mathbf{X}'_l) \tag{1}$$

$$\mathbf{X}'_{l+1} = \mathrm{LN}(\mathrm{FFN}(\mathbf{Y}_l) + \mathbf{Y}_l) \tag{2}$$

where $\mathbf{Y}_l$ denotes the residual output of the first sub-layer (Multi-Head Attention, abbreviated as MHA) and the layer input $\mathbf{X}'_l$. $\mathbf{X}'_{l+1}$ denotes the layer output. While PostNorm often yields strong model performance and regularization, it is difficult to train in very deep networks. The gradients can vanish or explode, necessitating careful learning rate warm-up and initialization strategies.

**Pre-LayerNorm (PreNorm):** To address the training instability of PostNorm, especially when training deep models, the PreNorm variant was proposed. Here, LayerNorm is applied to the input of each sub-layer, before the residual connection. This creates a clean, identity-based residual path. The computation flow is:

$$\mathbf{Y}_l = \mathrm{MHA}(\mathrm{LN}(\mathbf{X}'_l)) + \mathbf{X}'_l \tag{3}$$

$$\mathbf{X}'_{l+1} = \mathrm{FFN}(\mathrm{LN}(\mathbf{Y}_l)) + \mathbf{Y}_l \tag{4}$$

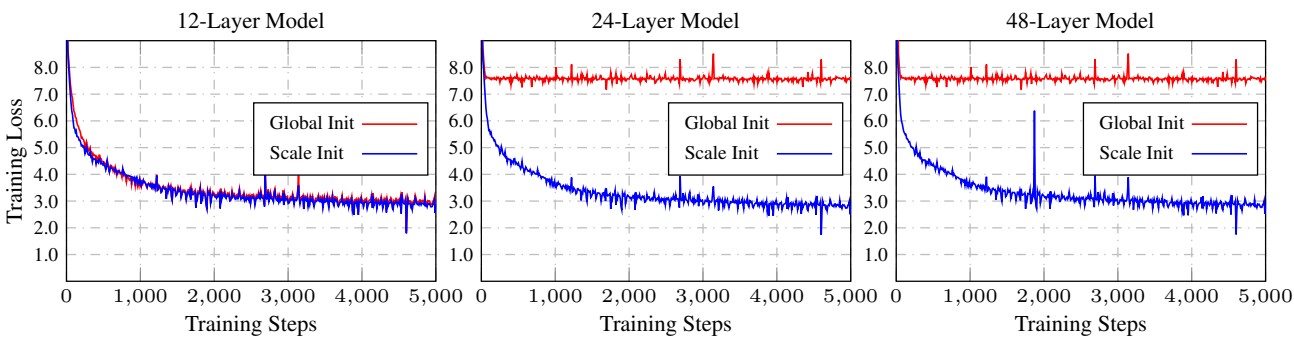

*Figure 2.* Early-stage training stability analysis. We train dense models with a fixed hidden dimension $d = 1536$ across increasing depths (12, 24, 48 layers) for 5000 steps to evaluate the impact of initialization on stability.

## 3.2. The SpanNorm Architecture Design

The goal of SpanNorm is to combine the performance benefits of PostNorm with a structural design that promotes the training stability characteristic of PreNorm. We achieve this by modifying the information flow in the second residual connection of a PostNorm block.

Given the input to the block $X'_l$, the computation proceeds as follows:

$$\mathbf{Y}_l = \text{LN}(\text{MHA}(\mathbf{X}'_l) + \mathbf{X}'_l) \tag{5}$$

$$\mathbf{X}'_{l+1} = \text{LN}(\text{FFN}(\mathbf{Y}_l) + \mathbf{X}'_l) \tag{6}$$

and the pseudocode is shown in Algorithm 1. The first sublayer (i.e., MHA) follows the standard PostNorm structure. However, in the second sub-layer (Feed-Forward Network, abbreviated as FFN), the residual connection adds the original block input, $\mathbf{X}'_l$, instead of the intermediate representation, $\mathbf{Y}_l$. We term this SpanNorm because the residual connection effectively spans the entire transformation block. This minor change creates a powerful direct skip connection from the input to the output of the entire block.

We make a specific modification for the first layer of the network ($l = 1$). The input to this layer consists of raw word embeddings, $\mathbf{E}$, which are not normalized, unlike the inputs to subsequent layers. To ensure the MHA sub-layer operates on a normalized input and maintains a consistent attention pattern across the network, we apply an additional LayerNorm to the embeddings before the MHA computation. However, this initial LayerNorm is not applied to $\mathbf{E}$ within the identity path of the residual connection, thereby preserving a clean, unmodified "highway" for information to flow seamlessly. Thus, the forward pass for the first layer is defined as:

$$\mathbf{Y}_1 = \text{LN}\left(\text{MHA}(\text{LN}(\mathbf{E})) + \mathbf{E}\right) \tag{7}$$

$$\mathbf{X}'_2 = \text{LN}\left(\text{FFN}(\mathbf{Y}_1) + \mathbf{E}\right) \tag{8}$$

## 3.3. Comparative Analysis of Gradient Dynamics

The design of SpanNorm strategically combines the structural strengths of PreNorm and PostNorm. In this section, we provide a theoretical analysis of gradient dynamics, demonstrating how SpanNorm mitigates the specific pathologies of its predecessors: (1) the representation collapse characteristic of deep PreNorm networks, and (2) the vanishing gradients inherent to PostNorm architectures.

**Advantage over PreNorm: Preventing Deep-Layer Representation Collapse** A primary pathology in deep PreNorm Transformers is the uncontrolled growth of feature variance, which leads to representation collapse. In a standard PreNorm block, the main residual path acts as an identity map, causing variance to accumulate linearly with depth. Formally, for a network of depth $L$, the variance of the hidden states scales as $\text{Var}(X'_L) = \Theta(L)$ (Kedia et al., 2024).

This variance explosion degrades the learning capability of deep layers. Consider the Jacobian of the $l$-th PreNorm block. Let $\text{Res}(\cdot)$ denote the transformation within the residual branch (e.g., Self-Attention or FFN). Since the transformation within the residual branch operates on inputs normalized by the feature standard deviation $\sigma_l = \sqrt{\text{Var}(X'_l)}$, the Jacobian $J_{\text{Pre}}$ can be expressed as:

$$J_{\text{Pre}} = \mathbb{I} + \underbrace{\frac{\partial \text{Res}(X'_l)}{\partial X'_l} \cdot \mathcal{O}\left(\frac{1}{\sigma_l}\right)}_{\text{Vanishing Term}} \tag{9}$$

As $l \to \infty$, $\sigma_l \to \infty$, causing the residual term to vanish at a rate of $\mathcal{O}(1/\sqrt{l})$. Consequently, $J_{\text{Pre}} \to \mathbb{I}$, meaning the gradients effectively bypass the transformative sub-layers. The deep layers thus devolve into identity functions, contributing minimal new information to the representation.

*The SpanNorm Solution.* SpanNorm circumvents this issue by enforcing a variance reset at the end of each block (Eq. 6). By applying LayerNorm to the combined output of the FFN and the skip connection, SpanNorm ensures that the

output variance remains bounded, i.e., $\text{Var}(X'_{l+1}) = \Theta(1)$. This prevents the Jacobian collapse, forcing deep layers to contribute meaningfully to the feature transformation.

**Advantage over PostNorm: Enhancing Stability by Mitigating Gradient Decay.** While standard PostNorm models maintain bounded variance, they suffer from exponential gradient decay due to the serial application of normalization layers on the main propagation path. To formalize the advantage of SpanNorm, we analyze the signal attenuation factor under a simplified homogeneous setting.

**Assumption 3.1** (Homogeneous Variance). We assume that the pre-normalized sums in each sub-layer exhibit a consistent standard deviation $\sigma > 1$ across the network.

Under Assumption 3.1, the gradient magnitude $\|G\|$ passing through a normalization layer is scaled by a factor of $1/\sigma$. We compare the cumulative scaling behavior at depth $L$:

*PostNorm Decay.* In a PostNorm block, the signal passes through two normalization layers sequentially (one after Attention, one after FFN). The gradient norm at layer $L$ is proportional to the product of these scaling factors:

$$\|G_{\text{Post}}(L)\| \propto \left(\frac{1}{\sigma} \cdot \frac{1}{\sigma}\right)^L = \sigma^{-2L} \tag{10}$$

*SpanNorm Preservation.* In contrast, SpanNorm's topology (Eq. 6) introduces a single normalization on the aggregate block output $X'_{l+1}$. The gradient signal is effectively scaled only once per block regarding the macro-architecture: $\|G_{\text{Span}}(L)\| \propto \left(\frac{1}{\sigma}\right)^L = \sigma^{-L}$.

Comparing the decay rates reveals a fundamental advantage:

$$\|G_{\text{Span}}(L)\| \propto \sqrt{\|G_{\text{Post}}(L)\|} \tag{11}$$

In the regime of deep networks where gradients vanish ($\|G\| \ll 1$), this square-root relationship represents a qualitative shift in stability. It converts the quadratic decay mechanism of PostNorm ($\sigma^{-2}$ per layer) into a much gentler linear decay ($\sigma^{-1}$), preventing the vanishing gradient problem at its root. For deep networks (e.g., $L = 128$) with $\sigma \approx 1.05$, this theoretical difference results in gradient signals for SpanNorm that are orders of magnitude larger than those of PostNorm ($1.05^{128} \approx 500\times$), effectively bridging the gap between divergence and convergence.

# 4. Principled Initialization for Depth Scalability

> **Recipe:** Scale the initialization variance of the FFN and Attention output matrices by $1/L$.

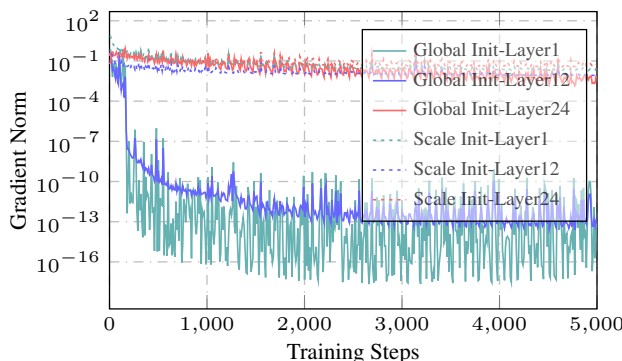

*Figure 3.* Gradient norm dynamics (Analysis of the 24-layer model in Figure 2). We visualize layer-wise gradient norms for the exact 24-layer configuration that exhibited instability. (See Figure 14 for the full visualization of all 24 layers).

To ensure stable training as network depth increases, we analyze the conditions for stable initialization. The stability is governed by the spectral norm of the layer-wise Jacobian matrix, which must remain close to 1 to prevent exploding or vanishing gradients. Our analysis of the proposed SpanNorm architecture is summarized in the following theorem.

**Theorem 4.1** (Condition for Stable Depth Scaling). *For a SpanNorm Transformer model with $L$ layers to maintain stable training dynamics as $L \to \infty$, the output variance of its residual sub-layers (e.g., the FFN block $\mathcal{F}$) must scale inversely with the total depth. Specifically, the condition is:*

$$\text{Var}(\mathcal{F}(\text{LN}(\mathbf{x}))) = \mathcal{O}(1/L) \tag{12}$$

**Practical Initialization Strategy.** Theorem 4.1 offers a clear initialization strategy. To directly meet the variance condition $\text{Var}(\mathcal{F}(\cdot)) = \mathcal{O}(1/L)$, initializing the FFN block parameters is crucial. We recommend setting the standard deviation of the output matrix $\mathbf{W}_2$'s initialization to $\mathcal{O}(1/\sqrt{L})$, which effectively reduces the matrix norm and ensures the FFN's output variance aligns with $\mathcal{O}(1/L)$. Similarly, the attention output matrix $\mathbf{W}_O$ should be scaled to preserve gradient integrity within the FFN block and support the near-identity principle of deep residual networks. Thus, we apply $\mathcal{O}(1/\sqrt{L})$ scaling to both $\mathbf{W}_2$ and $\mathbf{W}_O$. Notably, this theoretical approach matches the initialization method used in frameworks like Megatron-LM (Shoeybi et al., 2019). While we apply this initialization globally to ensure fairness, our experiments in Section 5 demonstrate that PostNorm still fails to converge at scale, validating our analysis in Section 3.3 that the superior stability of SpanNorm stems from its architectural topology rather than specific initialization strategies.

**Experimental Validation.** To empirically validate the effectiveness of the scaled initialization, we conducted exper-

*Table 1.* Our SpanNorm results against PreNorm (Touvron et al., 2023) on various configurations. All models are trained on the same subset of the SlimPajama dataset (from 30B to 200B) with the Mistral tokenizer (Jiang et al., 2023). The last column shows the average over all benchmarks that use (normalized) accuracy as the metric.

| Model | Param | Tokens | Wiki. ppl↓ | LMB. ppl↓ | LMB. acc↑ | PIQA acc↑ | Hella. acc_norm↑ | SciQ acc↑ | ARC-c acc_norm↑ | Wino. acc↑ | Avg. acc↑ |
|---|---|---|---|---|---|---|---|---|---|---|---|
| *Dense Models* | | | | | | | | | | | |
| PreNorm | 740M | 30B | 22.5 | 22.9 | 39.5 | 66.3 | 39.3 | 79.1 | 25.3 | 49.8 | 49.8 |
| SpanNorm | 740M | 30B | 20.6 | 26.3 | 38.5 | 67.9 | 42.4 | 80.1 | 25.1 | 50.8 | 50.8 |
| PreNorm | 1.3B | 100B | 17.4 | 12.6 | 51.1 | 70.0 | 49.6 | 84.4 | 26.8 | 53.1 | 55.8 |
| SpanNorm | 1.3B | 100B | 15.7 | 12.0 | 52.1 | 72.3 | 54.0 | 85.5 | 28.8 | 55.3 | 58.0 |
| PreNorm | 5B | 200B | 12.5 | 6.9 | 61.4 | 73.7 | 64.4 | 90.7 | 34.4 | 60.1 | 64.1 |
| SpanNorm | 5B | 200B | **11.8** | **5.8** | **64.2** | **75.7** | **66.9** | **92.0** | **36.3** | **64.2** | **66.6** |
| *MoE Models* | | | | | | | | | | | |
| PreNorm | A2.4B-16B | 200B | 12.0 | 6.7 | 61.7 | 75.9 | 66.0 | 89.1 | 37.0 | 60.9 | 65.1 |
| SpanNorm | A2.4B-16B | 200B | **11.4** | **6.5** | **63.0** | **76.2** | **68.1** | **90.8** | **38.6** | **61.3** | **66.3** |

*Table 2.* Comparisons with other advanced LN variants.

| Model | Param | Tokens | Wiki. ppl↓ | LMB. ppl↓ | LMB. acc↑ | PIQA acc↑ | Hella. acc_norm↑ | SciQ acc↑ | ARC-c acc_norm↑ | Wino. acc↑ | Avg. acc↑ |
|---|---|---|---|---|---|---|---|---|---|---|---|
| PostNorm | 5B | 200B | | | | | failed | | | | |
| PreNorm | 5B | 200B | 12.5 | 6.9 | 61.4 | 73.7 | 64.4 | 90.7 | 34.4 | 60.1 | 64.1 |
| HybridNorm | 5B | 200B | 12.2 | 6.5 | 62.7 | 75.1 | 66.2 | 91.4 | 35.4 | 61.3 | 65.4 |
| Mix-LN | 5B | 200B | 12.3 | 6.9 | 61.3 | 75.6 | 65.1 | 90.1 | 34.1 | 61.4 | 64.6 |
| LayerNorm Scaling | 5B | 200B | 13.6 | 14.9 | 51.0 | 73.1 | 60.7 | 86.1 | 32.2 | 58.3 | 60.2 |
| Peri-LN | 5B | 200B | 12.3 | 7.2 | 59.5 | 75.7 | 65.1 | 90.3 | 35.2 | 59.9 | 64.3 |
| SpanNorm | 5B | 200B | **11.8** | **5.8** | **64.2** | **75.7** | **66.9** | **92.0** | **36.3** | **64.2** | **66.6** |

iments on models with 12, 24, and 48 layers, as shown in Figure 2. We observed that a standard global initialization leads to training collapse when scaled to 24 layers, while the scaled initialization allows stable training even at 48 layers. A closer inspection of the gradient dynamics for the failing 24-layer configuration (Figure 3) confirms that the instability stems from vanishing gradients in the initial layers. By mitigating this issue, our scaled initialization ensures that valid gradient signals propagate to the bottom of the network, which is essential for training deep Transformers.

## 5. Experiments

**Experimental Setups** We summarize the detailed setups of selected baselines, model configurations, training hyperparameters and the corresponding dataset and evaluation protocols in Appendix B.

**Results on LM Evaluation Harness** We conducted a series of experiments across various model sizes, utilizing both dense and MoE architectures. The comprehensive results, presented in Table 1, demonstrate a clear and consistent performance advantage for our approach when compared to the PreNorm baseline. Specifically, our method yields an average improvement of 1.0-2.5 points across different dense model capacities (from 740M to 5B). Similarly, our

SpanNorm can also achieve an average improvement of 1.2 when switching to the MoE setting, further demonstrating its effectiveness. Notably, our design also addresses a common optimization challenge by enabling stable training of PostNorm models (which failed to converge), effectively eliminating the gradient vanishing.

**Comparisons with Other LayerNorm Variants** To comprehensively understand the advantages of SpanNorm, we also compared it with several recent variants, as shown in Table 2. To ensure a fair comparison, we re-implemented these methods, including Mix-LN (Li et al., 2024), Hybrid-Norm (Zhuo et al., 2025), LayerNorm Scaling (Sun et al., 2025) and Peri-LN (Kim et al., 2025) in our codebase using Megatron and conducted experiments on a 5B dense model trained on 200B tokens for robust results. We summarize the results of the LM Harness evaluation in Table 2 and plot the training curves in Figure 4 (left). Unexpectedly, LayerNorm Scaling yielded worse results than the PreNorm baseline with the same hyperparameters. For LayerNorm Scaling, we attribute this phenomenon to its scaling factor, which is strongly dependent on the total depth of the model. In models with a large depth (e.g., 64), the gradients of shallower layers may be scaled down too much, which negatively impacts the optimization process. Meanwhile, Peri-LN, HybridNorm and Mix-LN (PostNorm ratio

*Table 3.* Deep scaling evaluation: Comparisons with advanced LN variants on a 128-layer model.

| Model | Param | Tokens | Wiki. ppl ↓ | LMB. ppl ↓ | LMB. acc ↑ | PIQA acc ↑ | Hella. acc_norm ↑ | SciQ acc ↑ | ARC-c acc_norm ↑ | Wino. acc ↑ | Avg. acc ↑ |
|---|---|---|---|---|---|---|---|---|---|---|---|
| PostNorm | 6.5B | 400B | | | | | failed | | | | |
| PreNorm | 6.5B | 400B | 11.9 | 6.6 | 62.6 | 75.3 | 65.2 | 90.5 | 32.6 | 60.1 | 64.4 |
| HybridNorm | 6.5B | 400B | | | | | failed | | | | |
| Mix-LN | 6.5B | 400B | 11.4 | 6.2 | 64.0 | 76.1 | 67.5 | 90.4 | 35.7 | 62.5 | 66.0 |
| SpanNorm | 6.5B | 400B | **10.4** | **5.5** | **65.4** | **77.5** | **70.6** | **92.6** | **37.7** | **66.6** | **68.4** |

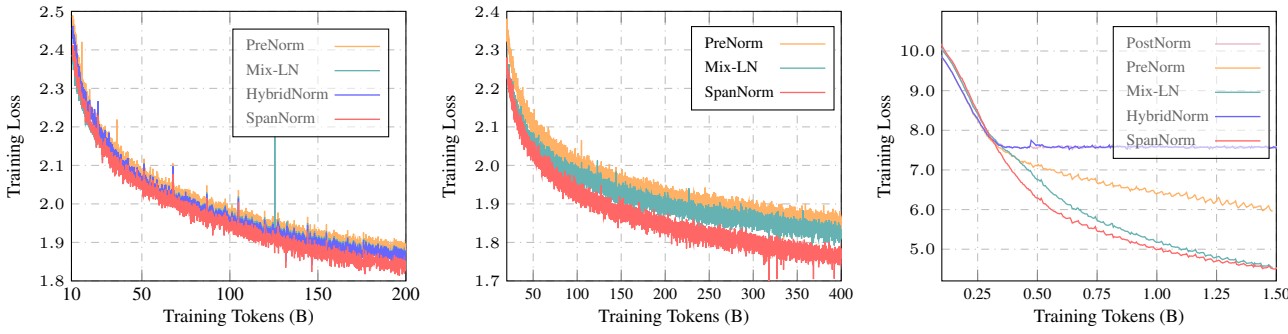

*Figure 4.* Performance and scalability across scales. (Left) SpanNorm consistently outperforms baselines on a 64-layer 5B dense model (200B tokens). (Middle & Right) On an ultra-deep 128-layer 6.5B model (400B tokens), SpanNorm achieves superior performance with robust convergence, whereas PostNorm and HybridNorm suffer from catastrophic divergence in the early phase (Right).

$\alpha = 25\%$) deliver better performance than PreNorm, and our SpanNorm consistently outperforms these advanced variants in both training loss and downstream performance.

**Scalability on Ultra-Deep Networks.** We stress-test stability by scaling to 128 layers (6.5B parameters). Table 3 and Figure 4 focuses exclusively on the most competitive variants from Table 2, excluding those that already underperformed the PreNorm. Our results reveal that success at moderate depths does not guarantee scalability. Notably, despite its effectiveness in shallower settings, HybridNorm suffered from optimization divergence in the deep regime. In contrast, SpanNorm demonstrated robust stability, not only converging successfully but also outperforming PreNorm by a significant margin (achieving an average improvement of up to 4.0). This validates SpanNorm's unique capacity for "stable transfer", preventing catastrophic training collapses in industrial-scale scaling.

## 6. Empirical Analysis

In this section, we provide empirical validation for the theoretical advantages of our SpanNorm architecture discussed in Section 3.3.

**Empirical Validation of Preventing Representation Collapse.** To empirically validate the issue of representation collapse in PreNorm and assess whether SpanNorm can mitigate this problem, we examine the evolution of its in-

ternal representations. The gradient analysis suggests that the Jacobian of a deep PreNorm block converges to an identity matrix, which implies that deeper layers fail to learn new features, causing the output of one layer to approximate the next. Figure 5 presents the cosine similarity between the hidden state outputs across all layer pairs in the MoE-A2.4B-16B model. This visualization reveals that the contrast between the architectures is most pronounced in deeper layers: while PreNorm retains high similarity, SpanNorm evolves towards markedly distinct representations. To confirm this observation quantitatively, we aggregate the similarity scores by layer distance in Figure 6, calculating the mean cosine similarity for all layer pairs $(L_i, L_j)$ separated by a distance $k = |i - j|$. The results show that PreNorm maintains a stagnant, flat curve ($> 0.5$) across all distances, whereas SpanNorm exhibits a rapid, near-linear decay down to $\approx 0.25$. These results demonstrate that SpanNorm effectively drives long-range feature evolution, ensuring that deep layers contribute meaningful information unlike the redundant layers in PreNorm.

**Empirical Validation of Mitigated Gradient Decay.** We empirically validate the severe gradient decay issue in PostNorm and examine whether SpanNorm can alleviate this problem. Our experiments are conducted on a 24-layer model with 740 million parameters. A comparison with a standard PostNorm model across three learning rates demonstrates the distinct advantages of SpanNorm, as illustrated in Figure 7. At a low learning rate ($8 \times 10^{-5}$), SpanNorm

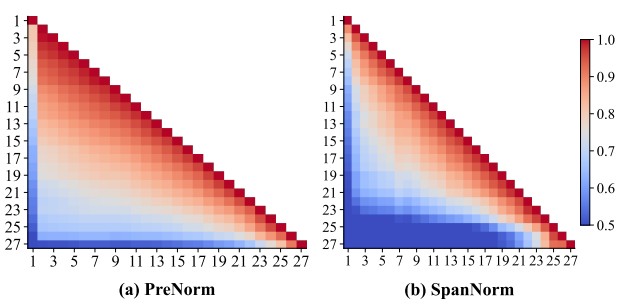

*Figure 5.* Cosine similarity between the outputs of each pair of layers for MoE-A2.4B-16B model.

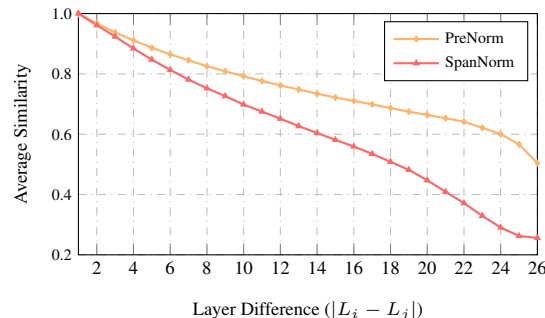

*Figure 6.* Quantitative analysis of representation collapse (Complementing Figure 5).

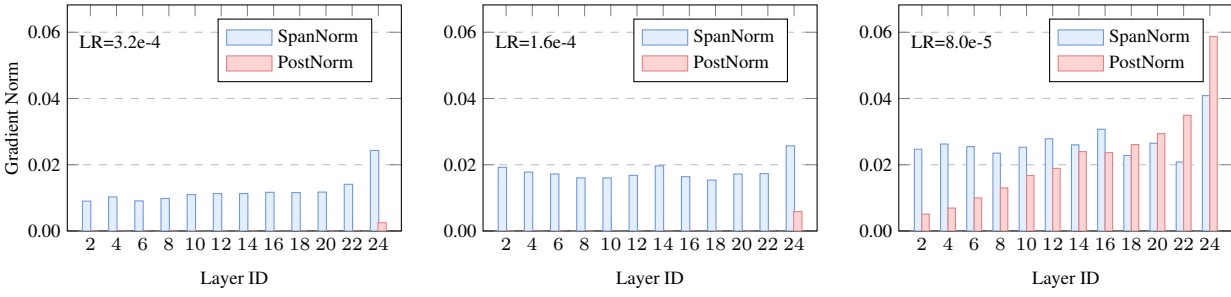

*Figure 7.* Comparison of FFN weight gradient norms between our SpanNorm model and standard PostNorm using the 740M model configuration (detailed in Appendix B) across three learning rate conditions. (Left and Middle) At a high learning rate ($> 8 \times 10^{-5}$), PostNorm training collapses, showing near-zero gradients in all but the final layer, while SpanNorm remains stable. (Right) At a lower learning rate ($8 \times 10^{-5}$), PostNorm suffers significant shallow-layer gradient decay, whereas SpanNorm maintains balanced gradients across all layers, demonstrating superior training stability and healthier gradient propagation.

sustains well-balanced gradients in contrast to the skewed distribution observed in PostNorm, which indicates pronounced shallow-layer decay. As the learning rate increases, PostNorm undergoes catastrophic collapse, whereas SpanNorm maintains stability. Notably, SpanNorm consistently exhibits a balanced gradient distribution with minimal variance between layers across all learning rate settings. This empirical evidence suggests that the healthy gradient propagation achieved by SpanNorm contributes to more robust training dynamics and extends the stable learning rate range.

**Empirical Validation of Spectral Expressivity.** Our analysis directly quantifies SpanNorm's enhanced spectral expressivity (as detailed in Appendix C). First, regarding capacity, SpanNorm achieves an Effective Dimension Ratio (Jha & Reagen, 2025) of 30.66, more than doubling PreNorm's 12.99 (as shown in Figure 9). This substantial gap indicates that SpanNorm effectively mitigates representation collapse, ensuring that the model utilizes a much larger portion of its latent capacity to encode diverse features, whereas PreNorm is confined to a low-rank subspace. Second, the Eigenvalue Distribution (Figure 10) analysis highlights the geometric nature of this advantage. Unlike PreNorm, which suffers from a sharp singular value decline and collapses into a low-dimensional subspace, SpanNorm

preserves a flatter, heavy-tailed spectrum. This slower decay ensures the utilization of diverse eigen-directions, preventing the spectral dilution characteristic of deep PreNorm models. Finally, regarding stability, SpanNorm maintains significantly lower condition numbers (Figure 11), mitigating the ill-conditioning of deep PreNorm layers. This ensures numerical stability and prevents matrix degeneracy.

**Stress-Testing "Deeper is Better" at Extreme Depths.** To isolate the impact of depth while keeping the experiment tractable, we construct dense proxy models with a fixed hidden size $d = 384$ and vary only the number of layers, setting $L \in \{32, 64, 128, 256, 512\}$. All models use the same learning rate, $2 \times 10^{-4}$, together with Scale Init, without depth-specific tuning. This fixed-hyperparameter setup makes depth the primary variable and directly tests robustness to extreme scaling, since unstable architectures typically require increasingly conservative learning rates or additional tuning as depth grows. Deep PreNorm Transformers often suffer from "depth degradation," where adding layers yields diminishing returns (Sun et al., 2025). In contrast, Figure 8 shows that SpanNorm converges up to 512 layers and follows a monotonic "deeper is better" trend as depth doubles. This behavior suggests two complementary benefits of our design. First, SpanNorm combined with Scale

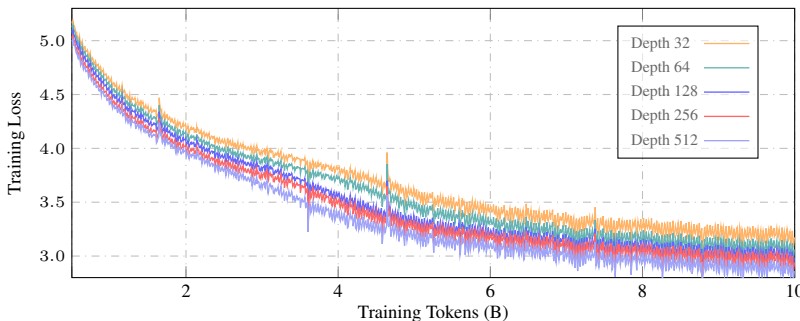

*Figure 8.* Stress-testing "Deeper is Better". We train proxy models ($d = 384$) with depths scaling exponentially from 32 to 512 using a fixed learning rate. SpanNorm shows a strict monotonic decrease in training loss, confirming robust stability and the effective avoidance of depth degradation at extreme scales.

*Table 4.* Empirical comparison of topological variants on the 5B Dense Model trained for 200B tokens.

| Model | Param | Tokens | Wiki. ppl ↓ | LMB. ppl ↓ | LMB. acc ↑ | PIQA acc ↑ | Hella. acc_norm ↑ | SciQ acc ↑ | ARC-c acc_norm ↑ | Wino. acc ↑ | Avg. acc ↑ |
|---|---|---|---|---|---|---|---|---|---|---|---|
| Parallel Transformer | 5B | 200B | 12.3 | 6.7 | 61.3 | 75.4 | 65.1 | 90.4 | 35.1 | 62.7 | 65.0 |
| ResiDual | 5B | 200B | 12.8 | 7.6 | 59.5 | 74.1 | 63.2 | 89.1 | 33.7 | 61.3 | 63.4 |
| SpanNorm | 5B | 200B | **11.8** | **5.8** | **64.2** | **75.7** | **66.9** | **92.0** | **36.3** | **64.2** | **66.6** |

Init stabilizes gradient propagation under PostNorm-style computation, avoiding the divergence commonly observed in very deep models. Second, the additional layers remain functionally useful: rather than collapsing toward identity mappings, they continue to contribute to optimization and improve training loss.

**Structural Topology Ablations** Beyond the in-place normalization adjustments evaluated in Section 5 (e.g., Hybrid-Norm, Peri-LN), another prominent line of research focuses on altering the information flow topology of the Transformer block to balance training stability and model performance such as Parallel Transformer (Chowdhery et al., 2023) and ResiDual (Xie et al., 2023). Parallel Transformers break the sequential execution of Attention and FFN to achieve higher kernel throughput. ResiDual introduces a dual-stream residual topology to balance gradient flow and feature expressivity. To validate these topological differences, we evaluate Parallel Transformers and Dual Residual architectures using the 5B parameter configuration trained on 200B tokens. As shown in Table 4, SpanNorm consistently outperforms both topological variants across all evaluated metrics, demonstrating that SpanNorm's approach of extending the residual span while preserving a single-stream sequential computation graph provides a structurally superior and more robust foundation for scaling.

**Ablations on GQA.** We further examine whether Span-Norm delivers consistent improvements with alternative attention variants. Since our MoE experiments already use MLA, we conduct an additional ablation by replacing MHA

with GQA. Specifically, we use the 740M configuration with 24 query heads and 8 key-value heads, trained on 30B tokens. As shown in Table 5, SpanNorm consistently outperforms the PreNorm baseline under GQA, achieving lower perplexity and higher average downstream accuracy. This suggests that the benefits of SpanNorm are not tied to a specific attention implementation.

## 7. Conclusions

In this work, we addressed the long-standing trade-off between training stability and model performance in Transformer architectures. We introduced SpanNorm, a novel normalization strategy that synergistically combines the stability of the PreNorm design with the performance benefits of the PostNorm computation. Our theoretical analysis confirmed that SpanNorm ensures stable signal propagation, enabling the training of exceptionally deep and large models without encountering the gradient issues that typically hinder PostNorm architectures. Empirically, SpanNorm consistently outperforms existing normalization methods across various model scales. By resolving a critical bottleneck in LLM training, our work paves the way for the development of more powerful, efficient, and scalable models.

## Impact Statement

This paper presents work whose goal is to advance the field of machine learning. There are many potential societal consequences of our work, none of which we feel must be specifically highlighted here.

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

# A. Theoretical Proofs and Empirical Justifications

## A.1. Proof of Theorem 4.1

*Proof.* The forward pass for the $l$-th layer of our SpanNorm architecture is given by:

$$\mathbf{Y}_l = \text{LN}\left(\text{MHA}(\mathbf{X}'_l) + \mathbf{X}'_l\right) \tag{13}$$

$$\mathbf{X}'_{l+1} = \text{LN}\left(\text{FFN}(\mathbf{Y}_l) + \mathbf{X}'_l\right) \tag{14}$$

The stability of backpropagation depends on the spectral norm of the Jacobian matrix, $\left\|\frac{\partial \mathbf{X}'_{l+1}}{\partial \mathbf{X}'_l}\right\|_2$. Let $\mathbf{Z}_l = \text{FFN}(\mathbf{Y}_l) + \mathbf{X}'_l$, $\mathbf{A}_l = \text{MHA}(\mathbf{X}'_l) + \mathbf{X}'_l$. By the chain rule and the submultiplicativity of spectral norms, we have:

$$\left\|\frac{\partial \mathbf{X}'_{l+1}}{\partial \mathbf{X}'_l}\right\|_2 \leq \left\|\frac{\partial \text{LN}(\mathbf{Z}_l)}{\partial \mathbf{Z}_l}\right\|_2 \left\|\frac{\partial \mathbf{Z}_l}{\partial \mathbf{X}'_l}\right\|_2 \tag{15}$$

The spectral norm of the Layer Normalization Jacobian is approximately the inverse of the standard deviation of its input, $\|\frac{\partial \text{LN}(\mathbf{Z}_l)}{\partial \mathbf{Z}_l}\|_2 \approx \frac{1}{\sigma_{\mathbf{Z}_l}}$. The second term can be bounded using the triangle inequality (where FFN is denoted as $\mathcal{F}$):

$$\left\|\frac{\partial \mathbf{Z}_l}{\partial \mathbf{X}'_l}\right\|_2 = \left\|\mathbb{I} + \frac{\partial \mathcal{F}(\mathbf{Y}_l)}{\partial \mathbf{X}'_l}\right\|_2 \leq 1 + \left\|\frac{\partial \mathcal{F}(\mathbf{Y}_l)}{\partial \mathbf{X}'_l}\right\|_2 \tag{16}$$

The term $\left\|\frac{\partial \mathcal{F}(\mathbf{Y}_l)}{\partial \mathbf{X}'_l}\right\|_2$ can be further decomposed using the chain rule and submultiplicativity:

$$\left\|\frac{\partial \mathcal{F}(\mathbf{Y}_l)}{\partial \mathbf{X}'_l}\right\|_2 = \left\|\frac{\partial \mathcal{F}}{\partial \mathbf{Y}_l}\frac{\partial \mathbf{Y}_l}{\partial \mathbf{X}'_l}\right\|_2 \leq \left\|\frac{\partial \mathcal{F}}{\partial \mathbf{Y}_l}\right\|_2 \left\|\frac{\partial \mathbf{Y}_l}{\partial \mathbf{X}'_l}\right\|_2 \tag{17}$$

In the context of feedforward neural networks (FFNs), the Jacobian norm, represented as $\|\frac{\partial \mathcal{F}}{\partial \mathbf{Y}_l}\|_2$, can be approximated by the expression $\|\frac{\partial \mathcal{F}}{\partial \mathbf{Y}_l}\|_2 \approx \|\mathbf{W}_1 \mathbf{W}_2\|_2$. Here, $\mathbf{W}_1$ and $\mathbf{W}_2$ denote the weight matrices associated with the FFN's two linear layers, under the assumption of an identity activation function (Takase et al., 2023).

For the Jacobian of the first sub-layer, $\frac{\partial \mathbf{Y}_l}{\partial \mathbf{X}'_l}$, the formulation is more intricate. It can be bounded as $\|\frac{\partial \mathbf{Y}_l}{\partial \mathbf{X}'_l}\|_2 \approx \frac{1}{\sigma_{\mathbf{A}_l}}\|\mathbb{I} + \frac{\partial \text{MHA}}{\partial \mathbf{X}'_l}\|_2$. The Jacobian of the attention block, $\frac{\partial \text{MHA}}{\partial \mathbf{X}'_l}$, is further decomposed into the output projection matrix $\mathbf{W}_O$ and the preceding multi-head attention mechanism, which encompasses queries, keys, values, and the softmax operation. We denote the Jacobian of this attention mechanism as $\mathbf{J}^{\mathcal{A}'}$. Consequently, we have $\|\frac{\partial \text{MHA}(\mathbf{X}'_l)}{\partial \mathbf{X}'_l}\|_2 = \|\mathbf{W}_O \mathbf{J}^{\mathcal{A}'}\|_2$.

Combining these results, we obtain the final upper bound on the Jacobian's spectral norm:

$$\left\|\frac{\partial \mathbf{X}'_{l+1}}{\partial \mathbf{X}'_l}\right\|_2 \lesssim \frac{1}{\sigma_{\mathbf{Z}_l}}\left(1 + \frac{\|\mathbf{W}_2 \mathbf{W}_1\|_2(1 + \|\mathbf{W}_O \mathbf{J}^{\mathcal{A}'}\|_2)}{\sigma_{\mathbf{A}_l}}\right) \tag{18}$$

For stable training through $L$ layers, the cumulative product of these Jacobians must not vanish or explode. This is critically dependent on the outer scaling factor $\frac{1}{\sigma_{\mathbf{Z}_l}}$ at each layer. For the product to remain of order $\mathcal{O}(1)$ as $L \to \infty$, it is required that $\sigma_{\mathbf{Z}_l} = 1 + \mathcal{O}(1/L)$.

We analyze the variance of $\mathbf{Z}_l$. By the definition of LayerNorm, the input $\mathbf{X}'_l$ has unit variance, i.e., $\text{Var}(\mathbf{X}'_l) = 1$. The input to the FFN, $\mathbf{Y}_l$, is similarly normalized. Under the standard initialization assumption that the residual branch $\mathcal{F}(\mathbf{Y}_l)$ and the identity path $\mathbf{X}'_l$ are uncorrelated, the variance of their sum is the sum of their variances:

$$\text{Var}(\mathbf{Z}_l) = \text{Var}(\mathbf{X}'_l) + \text{Var}(\mathcal{F}(\mathbf{Y}_l)) = 1 + \text{Var}(\mathcal{F}(\mathbf{Y}_l)) \tag{19}$$

For the condition $\sigma_{\mathbf{Z}_l} = 1 + \mathcal{O}(1/L)$ to hold, the variance must be $\text{Var}(\mathbf{Z}_l) = (1 + \mathcal{O}(1/L))^2 = 1 + \mathcal{O}(1/L)$. Comparing this with our derived variance, we arrive at the central condition:

$$\text{Var}(\mathcal{F}(\mathbf{Y}_l)) = \mathcal{O}(1/L) \tag{20}$$

This completes the proof. $\qquad\square$

*Table 5.* Empirical comparison of SpanNormand PreNorm with GQA.

| Model | Param | Tokens | Wiki. ppl ↓ | LMB. ppl ↓ | LMB. acc ↑ | PIQA acc ↑ | Hella. acc_norm ↑ | SciQ acc ↑ | ARC-c acc_norm ↑ | Wino. acc ↑ | Avg. acc ↑ |
|---|---|---|---|---|---|---|---|---|---|---|---|
| PreNorm (w/ GQA) | 740M | 30B | 22.1 | 22.1 | 39.3 | 67.5 | 42.0 | 80.0 | 23.2 | **52.5** | 50.8 |
| SpanNorm (w/ GQA) | 740M | 30B | **20.4** | **21.5** | **40.5** | **68.3** | **43.5** | **82.1** | **24.5** | 52.1 | **51.8** |

## A.2. Empirical Justification of Assumption 3.1

To support the validity of the Homogeneous Variance assumption 3.1, we closely monitor the empirical statistics of the pre-normalized sums for both Attention ($\sigma_{A_l}$) and FFN ($\sigma_{Z_l}$) sub-layers. These metrics were tracked across all layers of our 64-layer 5B model configuration during actual training.

As summarized in Table 6, the empirical standard deviation $\sigma$ remains bounded and strictly greater than 1 across all network depths. Crucially, the ratio of $\sigma_{Z_l}$ to $\sigma_{A_l}$ remains tightly centered around 1.08, with values bounded between 0.96 and 1.19. This stable horizontal trajectory across all layers confirms that the sub-layer signals neither explode nor vanish during scaling. The empirical evidence demonstrates that deep layers operate under a homogeneous variance regime, fully validating the theoretical simplification adopted in our analysis.

*Table 6.* Layer-wise empirical standard deviation ratios $\sigma_{Z_l}/\sigma_{A_l}$ sampled across depths of the 64-layer 5B model.

| Layer ID | 4 | 8 | 12 | 16 | 20 | 24 | 28 | 32 |
|---|---|---|---|---|---|---|---|---|
| $\sigma_{Z_l}$ | 1.3302 | 1.1277 | 1.1543 | 1.2008 | 1.1771 | 1.2450 | 1.2154 | 1.2526 |
| $\sigma_{A_l}$ | 1.1487 | 1.1791 | 1.1496 | 1.1183 | 1.0968 | 1.1394 | 1.1197 | 1.1390 |
| **Ratio** | 1.16 | 0.96 | 1.00 | 1.07 | 1.07 | 1.09 | 1.09 | 1.10 |

| Layer ID | 36 | 40 | 44 | 48 | 52 | 56 | 60 | 64 |
|---|---|---|---|---|---|---|---|---|
| $\sigma_{Z_l}$ | 1.1063 | 1.1148 | 1.1481 | 1.3363 | 1.3185 | 1.2252 | 1.1950 | 1.3201 |
| $\sigma_{A_l}$ | 1.0963 | 1.0923 | 1.1061 | 1.1209 | 1.1163 | 1.1082 | 1.0549 | 1.2007 |
| **Ratio** | 1.01 | 1.02 | 1.04 | 1.19 | 1.18 | 1.11 | 1.13 | 1.10 |

## B. Detailed Experimental Setups

**Baseline** We evaluate our proposed SpanNorm on both dense and MoE settings. The PreNorm and PostNorm are two major baselines, and others like LayerNorm Scaling (Sun et al., 2025), HybridNorm (Zhuo et al., 2025), Mix-LN (Li et al., 2024), and Peri-LN (Kim et al., 2025) are stronger baselines that we have comprehensively compared.

**Model Configurations** We mainly evaluate our dense models across three configurations as follows: (1) 740M: The hidden dimension is 1536, and the intermediate size of FFN is 4224. The head is 24, with a head dimension of 64. The 740M is a 24-layer model. (2) 1.3B: The hidden dimension is 1536, and the intermediate size of FFN is 4224. The head is 24. We scaled the depth to twice that of the 740M model, which leads to a 48-layer model. (3) 5B: Our largest size model has a hidden dimension of 2560, an intermediate size of 6912, with 40 attention heads and a depth of 64. For the MoE model, we use the Deepseek-V3 small-scale model (Liu et al., 2024), activated by 2.4 billion parameters within approximately 16 billion parameters in total. It employs an MLA architecture, activating 6 out of 64 experts. The model consists of a total of 27 layers, with a hidden dimension of 2048.

**Training Hyperparameters** We employ the AdamW optimizer (Kingma & Ba, 2015) with hyper-parameters set to $\beta_1 = 0.9$ and $\beta_2 = 0.95$, with weight decay set to 0.1. We set the max sequence length to 2K during pretraining with 200B tokens uniformly sampled from SlimPajama. As for the learning rate schedule, we first linearly increase the learning rate from 0 to the peak value, e.g., $2 \times 10^{-4}$, during the first 200 steps. Then the learning rate decays following a cosine curve until meeting the minimum learning rate predefined. Our batch size is set to 512K for a 740M model, 1M for a 1.3B model, and 4M for a 5B model. The gradient clipping norm is set to 1.0. We chose the Megatron initialization (Shoeybi et al., 2019) (referred to as 'Scale Init' in Section 4) as our default strategy for all models, including the baselines (PostNorm, PreNorm, HybridNorm, etc.). This isolates the impact of the architecture from the initialization scheme.

To ensure a fair and rigorous comparison in the 5B dense model experiments in Table 2, we carefully calibrated the configurations for each model:

- Standard Learning Rate ($2 \times 10^{-4}$): We adopted this commonly used learning rate for the PreNorm baseline and SpanNorm. Additionally, LayerNorm Scaling and Mix-LN were also trained using this standard $2 \times 10^{-4}$ learning rate. Note that for Mix-LN, while the learning rate was kept standard, we performed a trade-off analysis on the architecture, finding that a configuration of 16 PostNorm layers and 48 PreNorm layers yielded the lowest training loss.

- Tuned Learning Rate ($1 \times 10^{-4}$): Certain baselines required lower learning rates for stability or optimal performance. For HybridNorm, we observed severe gradient vanishing with learning rates of $2 \times 10^{-4}$ and $1.5 \times 10^{-4}$; thus, we report the optimal stable result at $1 \times 10^{-4}$. Similarly, for Peri-LN, we empirically found that a $1 \times 10^{-4}$ learning rate yielded significantly better convergence and performance than $2 \times 10^{-4}$ for the Gemma-2 style architecture.

For the deep scaling experiments involving the 128-layer 6.5B dense model, to keep a fair comparison, we search for the best learning rate on PreNorm models, and set $1.2 \times 10^{-4}$ as the default setting for all other models (including Mix-LN, HybridNorm, PostNorm and SpanNorm).

Regarding the MoE experiments in Table 1, we utilized identical configurations for both the PreNorm baseline and our SpanNorm model, setting the learning rate to $2 \times 10^{-4}$.

**Dataset and Evaluation**   Following the setup in previous studies, we randomly sampled two distinct subsets, consisting of 200B and 400B tokens respectively, from the widely available open-source SlimPajama dataset. To evaluate the performance of LLMs, we report the results from the LM Harness evaluation, including Lambada, PIQA, HellaSwag, SciQ, ARC-c, and WinoGrande.

**PyTorch-Style Implementation of SpanNorm**   We provide a PyTorch-style implementation of SpanNorm below.

---

**Algorithm 1** Pseudocode for a Transformer block with SpanNorm

---

```python
# attn is one of attention implementations, it could be MHA, GQA, MLA, etc.
# q_proj, k_proj, v_proj are the projection layers for the query, key, and value.
# ffn is the feedforward layer, it could be replaced with MoE.

def forward(hidden_states):
    res = hidden_states

    # Attention block
    q = q_proj(hidden_states)
    k = k_proj(hidden_states)
    v = v_proj(hidden_states)
    hidden_states = attn(q, k, v)
    hidden_states = layer_norm(hidden_states + res)

    # FFN block
    hidden_states = ffn(hidden_states)
    hidden_states = layer_norm(hidden_states + res)

    return hidden_states
```

---

## C. Spectral Analysis of Feature Representations

### C.1. Analysis on Spectral Utilization

To quantify how effectively the FFN blocks utilize their high-dimensional latent space, we employ three diagnostic metrics: Hard Spectral Rank, Soft Spectral Rank, and the composite Effective Dimension Ratio (EDR) (Jha & Reagen, 2025), as visualized in Figure 9. We can observe that PreNorm suffers from severe spectral collapse, exhibiting the lowest

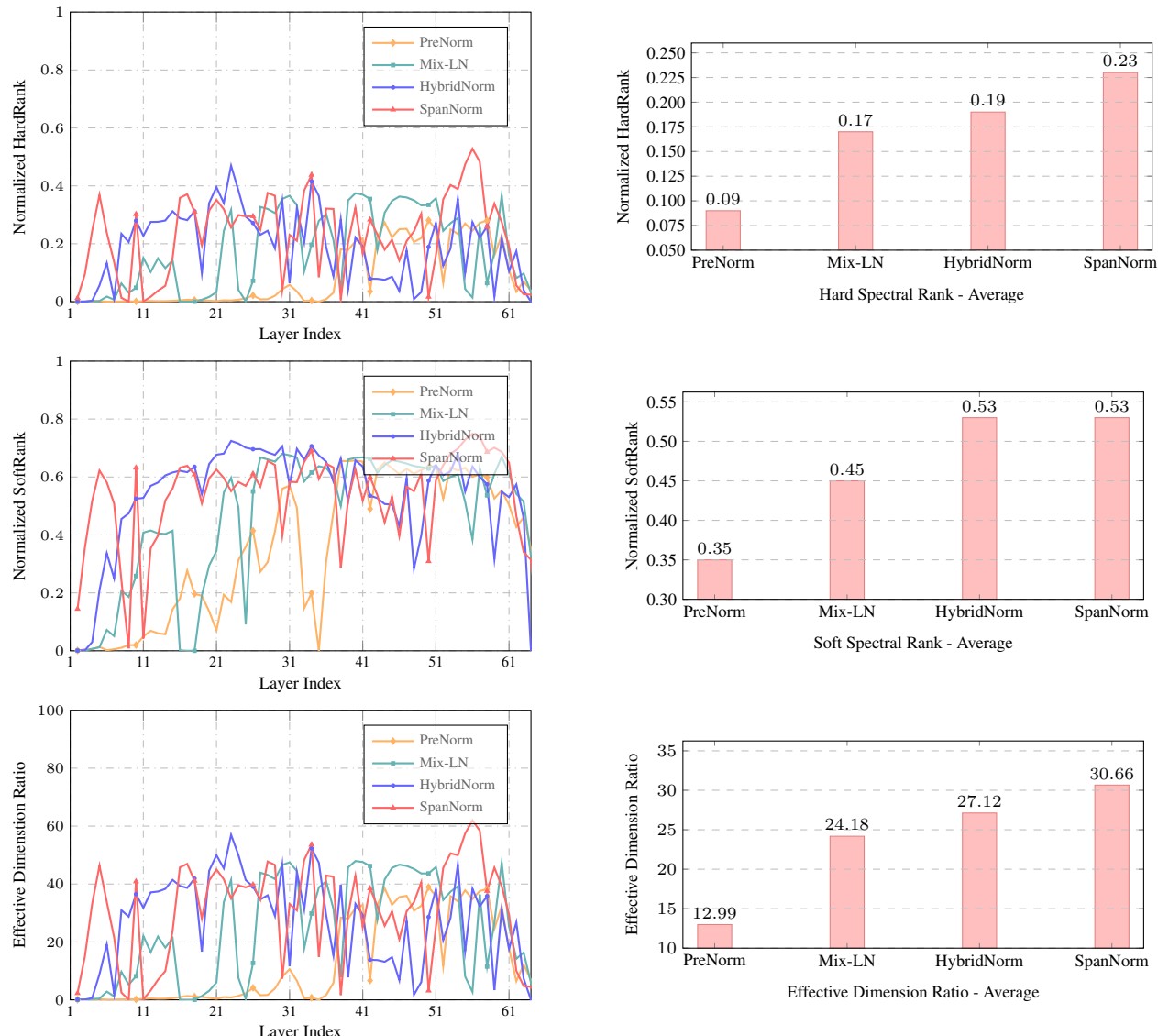

*Figure 9.* SpanNorm achieves superior spectral utilization compared to competitive normalization baselines. We evaluate the layer-wise dynamics (left) and average performance (right) of PreNorm, Mix-LN, HybridNorm, and SpanNorm across three spectral metrics. (Top) Hard Spectral Rank: SpanNorm demonstrates the strongest ability to preserve dominant eigen-directions, achieving the highest average rank of 0.23, significantly outperforming HybridNorm (0.19) and the PreNorm baseline (0.09). (Middle) Soft Spectral Rank: SpanNorm effectively prevents spectral dilution, tying for the top performance with a score of 0.53, ensuring a uniform distribution of information in the tail. (Bottom) Effective Dimension Ratio: Most notably, SpanNorm dominates in overall latent space efficiency with an effective ratio of 30.66, surpassing the second-best HybridNorm (27.12) and more than doubling the capacity of PreNorm (12.99). Overall, SpanNorm proves to be the most effective scheme for mitigating spectral collapse and maximizing representational expressivity.

utilization across all metrics (e.g., an average normalized HardRank of only 0.09). In contrast, SpanNorm demonstrates superior representational capability. For the dominant modes, SpanNorm achieves the highest average Hard Rank of 0.23, significantly outperforming the baselines. Regarding the spectral tail, SpanNorm ties for the top performance in Soft Rank (0.53), indicating it effectively prevents spectral dilution. Most notably, in terms of the Effective Dimension Ratio—which serves as a holistic measure of latent space efficiency—SpanNorm dominates with a score of 30.66, surpassing the second-best HybridNorm (27.12) and more than doubling the effective capacity of PreNorm (12.99). The layer-wise dynamics further reveal that SpanNorm maintains high and stable spectral utilization throughout the entire network depth. This empirical evidence demonstrates that SpanNorm successfully mitigates representational collapse, maximizing the expressivity of the learned features.

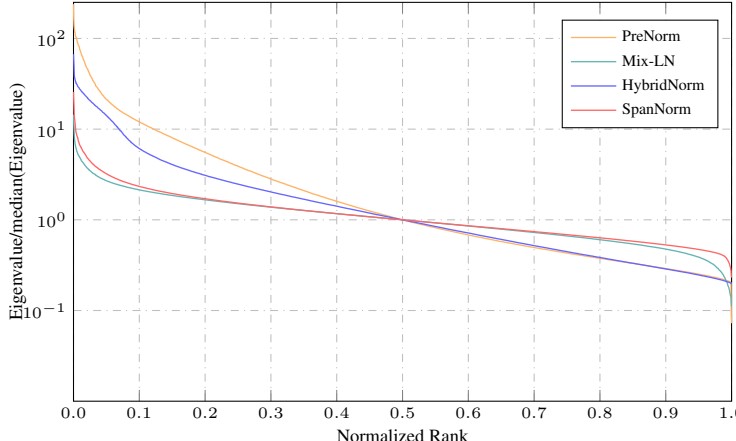

*Figure 10.* Eigenspectrum Analysis of Input Embeddings. We analyze the distribution of sorted eigenvalues normalized by their median value (log scale). While Mix-LN (teal) mitigates the extreme outliers seen in PreNorm, we observe that it suffers from a rapid decay in the tail spectrum, indicating potential dimensional collapse. In contrast, we demonstrate that SpanNorm (red) maintains a significantly flatter and more uniform trajectory across the entire rank, ensuring a more isotropic representation space with higher effective capacity.

### C.2. Analysis on Eigenvalue Distribution

Following the setup in Loshchilov et al. (2025)'s work, we mainly plot the sorted eigenvalues of the input embedding as shown in Figure 10. We can observe that SpanNorm exhibits a much flatter and more uniform spectrum across all ranks, while PreNorm and HybridNorm are dominated by a few large outliers and Mix-LN, although mitigating these outliers in the head, suffers from a rapid decay in the tail spectrum—indicating potential dimensional collapse. Moreover, we also plot the condition number of MLP1 (up-projection matrix) and MLP2 (down-projection matrix) at each layer. The left part in Figure 11 shows the condition numbers for MLP matrices at different layer depths, and the right part summarizes the corresponding average condition number. Our SpanNorm exhibits the smallest condition numbers in the MLP1 matrix. When switching to the MLP2, we see the condition number is the smallest across all 4 models at the bottom 6 and top 15 layers. The observation here demonstrates the ability of SpanNorm to preserve a well-conditioned and expressive latent space throughout the entire network depth.

Finally, we note that Nait Saada et al. (2025) investigated stable-rank collapse at initialization. However, their theory specifically targets bidirectional attention, which differs from our causal autoregressive setting. Nevertheless, our analysis shows that topological modifications outside the attention mechanism, like SpanNorm, do not alter this Softmax-driven static initialization gap. This clarifies the exact boundary of our method: SpanNorm is not designed to fix static rank collapse at initialization, but rather to resolve dynamic training pathologies, namely the gradient vanishing of PostNorm and the representation collapse of PreNorm, as network depth scales.

## D. Width Scaling and Predictability

While we prioritize depth scaling for structural stability, handling width scaling is equally critical for large-scale training. We empirically confirm that SpanNorm follows the width scaling protocols proposed by Everett et al. (2024). Crucially, this adherence facilitates the predictive stability transfer observed in our experiments.

As illustrated in Figure 12, we rigorously validate the transferability of stability thresholds across model widths. By fixing the depth and scaling the hidden dimension, we compare the training dynamics of a 310M proxy model ($d = 640$) against a 5B target model ($d = 2560$). The results confirm a robust predictive relationship: learning rates that fail on the proxy also fail on the target, while those stable on the proxy remain stable on the target. This stable transfer enables the use of efficient proxies to reliably identify safe hyperparameters, eliminating the need for risky tuning on large models. This is of paramount importance in industrial-scale model training, since failures incur prohibitive economic and temporal costs in industrial-scale training. To further assess the impact of this specific scaling strategy on downstream performance, we conducted a controlled ablation study on the 5B parameter dense model. Table 7 compares the standard PreNorm baseline against two SpanNorm configurations:

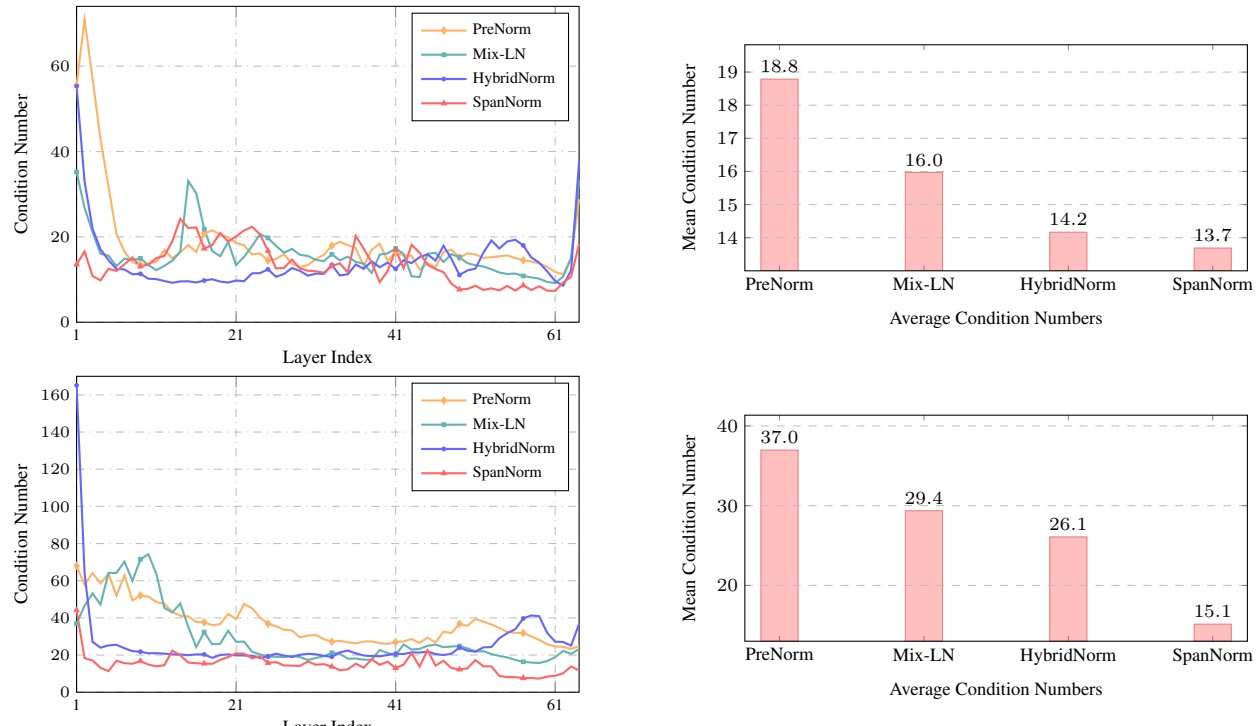

*Figure 11.* Layer-wise condition number analysis of FFN weights. We analyze the spectral properties of the MLP1 (up-projection) and MLP2 (down-projection) matrices across the network depth. SpanNorm demonstrates superior stability, maintaining significantly lower condition numbers, particularly in the deeper layers, which effectively prevents spectral degradation. In contrast, PreNorm exhibits elevated condition numbers in deeper blocks, reflecting the issue of deep layer degeneration where the representation capacity collapses. This comparison highlights SpanNorm's ability to preserve a well-conditioned and expressive latent space throughout the network depth.

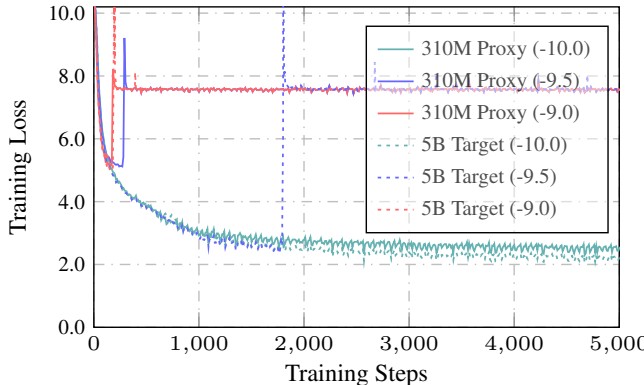

*Figure 12.* Validation of width scaling stability transfer. We compare a 310M proxy (hidden dimension $d = 640$) and a 5B target ($d = 2560$) under a fixed depth ($L = 64$).

- SpanNorm (w/o WS): Uses the exact same hyperparameters (e.g., learning rate) as the PreNorm baseline.

- SpanNorm (w/ WS): Applies our proposed Width Scaling strategy to adjust the learning rate based on the model width.

As shown in the table, the "w/o WS" setting yields slightly higher average accuracy (66.6 vs 65.9). We attribute this slight gap to the coarseness of the search grid used for the proxy model. The optimal learning rate identified on the proxy was constrained by a coarse discrete grid, meaning the transferred value ($2.4 \times 10^{-4}$) was an approximation rather than a precise optimum. A finer-grained search on the proxy would likely narrow this gap. Crucially, the Width Scaling strategy offers a predictable path for hyperparameter transfer, providing a safe baseline that guarantees stability without the need for expensive tuning on the target model.

*Table 7.* Ablation study of Width Scaling (WS) on the 5B Dense Model trained for 200B tokens. "w/o WS" denotes using the same hyperparameters as the PreNorm baseline, while "w/ WS" applies the width scaling strategy.

| Model | Param | Tokens | Wiki. ppl ↓ | LMB. ppl ↓ | LMB. acc ↑ | PIQA acc ↑ | Hella. acc_norm ↑ | SciQ acc ↑ | ARC-c acc_norm ↑ | Wino. acc ↑ | Avg. acc ↑ |
|---|---|---|---|---|---|---|---|---|---|---|---|
| PreNorm | 5B | 200B | 12.5 | 6.9 | 61.4 | 73.7 | 64.4 | 90.7 | 34.4 | 60.1 | 64.1 |
| SpanNorm (w/ WS) | 5B | 200B | **11.7** | 6.4 | 63.9 | **75.8** | **67.2** | 91.6 | 35.2 | 61.5 | 65.9 |
| SpanNorm (w/o WS) | 5B | 200B | 11.8 | **5.8** | **64.2** | 75.7 | 66.9 | **92.0** | **36.3** | **64.2** | **66.6** |

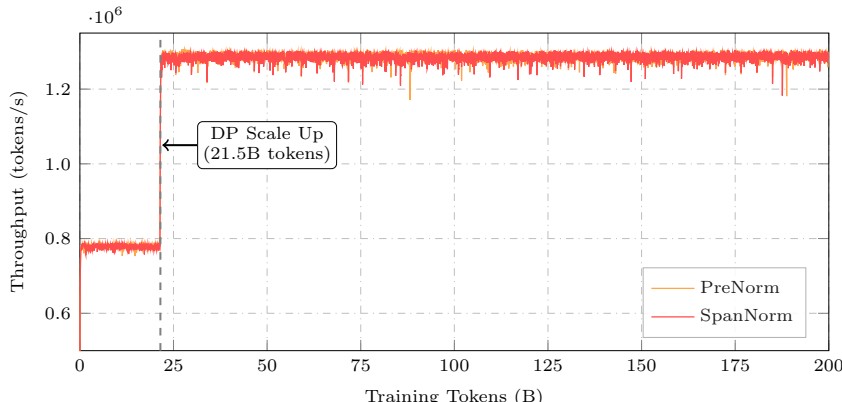

*Figure 13.* Training throughput record of the 5B model experiments. The plot compares the real-time throughput (tokens/s) of SpanNorm and PreNorm. The vertical dashed line marks a Data Parallelism (DP) scale-up event at 21.5B tokens, resulting in a step increase in throughput. SpanNorm maintains identical speed to the baseline both before and after the scaling, demonstrating consistent efficiency.

## E. Computational Efficiency Analysis

In this section, we empirically evaluate the training efficiency of SpanNorm compared to the standard PreNorm baseline. We recorded the real-time training throughput (measured in tokens per second) for the 5B parameter model experiments described in Section 5. As illustrated in Figure 13, the training speed of SpanNorm is virtually identical to that of the PreNorm baseline. The vertical dashed line in the figure marks a transition point at 21.5B tokens where we scaled up the Data Parallelism (DP) degree, resulting in a step increase in global throughput. Crucially, SpanNorm maintains consistent efficiency both before and after this scaling event. This result demonstrates that the architectural modifications and altered normalization placement in SpanNorm incur no computational overhead relative to the standard PreNorm baseline, making SpanNorm a practical drop-in replacement for large-scale training.

## F. Detailed Gradient Dynamics Analysis

In Section 4, we proposed a principled "Scale Init" strategy to mitigate the risk of gradient vanishing in deep networks. Figure 14 provides a comprehensive layer-wise visualization of the gradient norms for all 24 layers, offering a direct comparison between the standard Global Initialization and our proposed Scale Initialization. This detailed view confirms two critical observations:

- Collapse of Global Init: SpanNorm with standard Global Initialization (Red lines in bottom rows) exhibits catastrophic gradient decay, with norms dropping below $10^{-12}$, validating our theoretical analysis in Section 4.

- Stability of Scale Init: When paired with our proposed Scale Init (Blue lines in bottom rows), SpanNorm effectively maintains healthy gradient magnitudes ($\approx 10^{-1}$) across all depths, matching the stability profile of the PreNorm baseline.

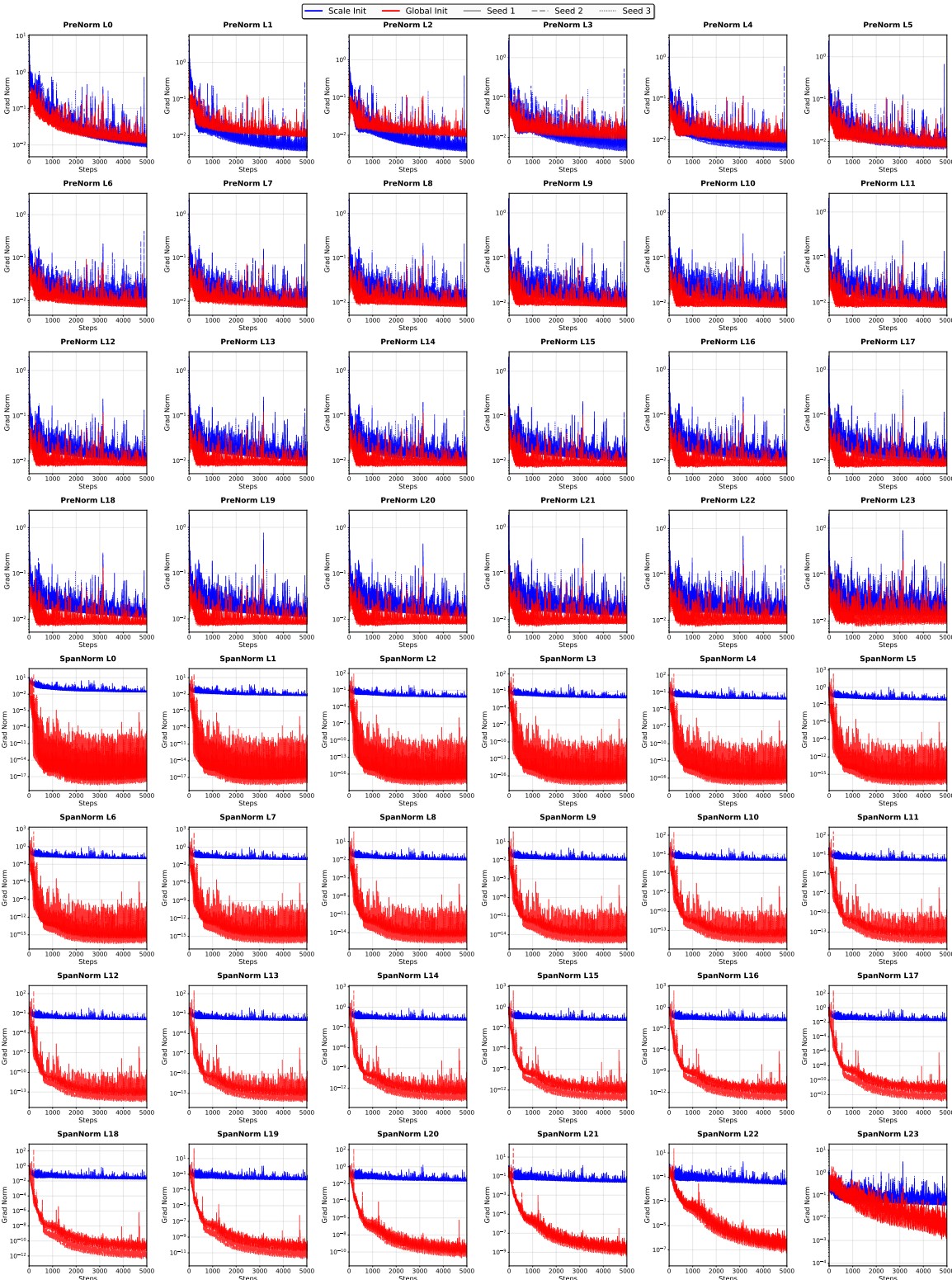

*Figure 14.* Comprehensive layer-wise gradient norm dynamics comparison (PreNorm vs. SpanNorm), which extends the original analysis of Figure 3 by visualizing gradient norms across all 24 layers for the first 5,000 training steps. We compare PreNorm (top rows) and SpanNorm (bottom rows) under two initialization strategies: the standard Global Init (Red) and our proposed Scale Init (Blue). The plots show the trend across 3 random seeds, with faint lines representing individual runs to demonstrate reproducibility. (1) Collapse without Scaling: The bottom rows reveal that SpanNorm with Global Init suffers from catastrophic gradient decay, where norms drop below $10^{-12}$, confirming the theoretical prediction of training collapse. (2) Effectiveness of Scale Init: Scale Init effectively stabilizes SpanNorm, maintaining healthy gradient norms ($\approx 10^{-1}$) comparable to the PreNorm baseline. (3) Baseline Comparison: PreNorm remains robust to initialization changes, but SpanNorm achieves similar stability only when paired with Scale Init.

