# OpenReview forum: "SpanNorm: Reconciling Training Stability and Performance in Deep Transformers"
_ICML.cc/2026/Conference — ICML 2026 regular_

### Official Review · Reviewer_EaQV · 2026-03-08

**Soundness:** 4
**Presentation:** 4
**Significance:** 4
**Originality:** 2
**Overall Recommendation:** 5
**Confidence:** 4

**Summary:**

The paper introduces SpanNorm, a novel architectural change to the transformer architecture. The change addresses the placement of layer normalization. The authors point out the dichotomy between "PreNorm" architectures, which provide stable training for deep networks but suffer from representation collapse in deeper models, and "PostNorm" architectures which suffer from training instability but provide strong performance.

The proposed architecture positions itself as a best-of-both-worlds: it has great training stability without sacrificing performance. It achieves this feat by establishing direct residual connections that spans the entire transformer block. Although it sounds similar to the trivial "sandwich" approach, where one would combine PreNorm and PostNorm, SpanNorm has the advantage of being computationally free.

Contributions of the paper:
* introducing SpanNorm, a novel architecture change to Transformers providing best of both worlds in terms if layer normalization: stable training and no performance degradation for deeper models.
* a principled discussion as to what makes SpanNorm efficient. The architecture change is elegant and well-motivated and an easy plug-and-play.
* Strong experimental results: tested on models of 6.5B parameters, up to 512 layers and deep mixture-of-experts.

**Compliance With Llm Reviewing Policy:**

Affirmed.

**Final Justification:**

I raised my score from 4 to 5. I think the paper had presented a solid intuition and they addressed my comments regarding the missing baselines. The results are solid and are run on a large scale.

The theoretical claims are a bit weaker than I initially thought however. So I am expecting the authors to better highlight the limitations in the final draft.

**Key Questions For Authors:**

Can the authors elaborate how their method compares against these two baselines? it would be ideal if you can also provide an empirical comparison (although I understand re-running experiments with models as big as the ones you use might not be feasible):

* Parallel transformers from [1]: Can you provide a justification as to why SpanNorm is better than Parallel Transformers? Parallel transformers have the advantage of being computionally faster while also spanning the residual connection back to the block input.
* ResiDual from [2]: How Does SpanNorm compare against this?

Thanks,


[1] Chowdhery, Aakanksha, et al. “PaLM: Scaling Language Modeling with Pathways.” arXiv.Org, 5 Oct. 2022, arxiv.org/abs/2204.02311.
[2] Xie, Shufang, et al. “ResiDual: Transformer with Dual Residual Connections.” arXiv.Org, 28 Apr. 2023, arxiv.org/abs/2304.14802. Accessed 07 Mar. 2026.

**Limitations:**

yes

**Strengths And Weaknesses:**

Strengths:
* Well-motivated and sound architecture. The paper establishes a coherent narrative as to why SpanNorm can provide strong performance while being stable in training. It basically combines both PreNorm and PostNorm while being computationally free. I believe the architectural change to be quite elegant.
* The paper provided good mathematical analysis of the gradient behavior (where PostNorm usually exhibits decaying gradient) and features' variance analysis (where PreNorm sees exploding variance).
* because the proposed change is plug-and-play, it would be an easy hanging-fruit to try for existing deep architectures. As such, the approach has great potential.
* Good execution of the experiments: training of 6.5B parameters model and 16B Mixture-of-Experts model.

Weaknesses:
* missing critical literature and baseline: the main novelty claim of the paper is establishing a clean residual connection back to the original block input. Parallel Transformers used in PaLM [1] also establishes a direct connection while also being 15% faster than regular layer normalization. I think this should have been clearly mentioned in the paper as I see as the direct competitor (and it already established itself in a massive model of 540B parameters). I also think it should have been part of the baseline.
* missing discussion of how the approach compares against another strong baseline: ResiDual [2].
 I think I need a detailed discussion of how the approach compares against these two baselines as they seem like direct competitors.



[1] Chowdhery, Aakanksha, et al. “PaLM: Scaling Language Modeling with Pathways.” arXiv.Org, 5 Oct. 2022, arxiv.org/abs/2204.02311.
[2] Xie, Shufang, et al. “ResiDual: Transformer with Dual Residual Connections.” arXiv.Org, 28 Apr. 2023, arxiv.org/abs/2304.14802. Accessed 07 Mar. 2026.

---

> ### Author Rebuttal · Authors · 2026-03-30
>
> We sincerely thank the reviewer for their positive assessment of our work. We are highly encouraged by your recognition of SpanNorm as a well-motivated, elegant, and computationally free "best-of-both-worlds" architecture. We also deeply appreciate your acknowledgment of our mathematical analysis and the rigorous scale of our empirical validation
>
> We completely agree that discussing and comparing against Parallel Transformers and ResiDual strengthens the paper.
>
> > Comparison with Parallel Transformers (PaLM)
>
> While Parallel architectures also establish a direct connection from the block input to the sub-layers, they possess two fundamental theoretical and structural differences compared to SpanNorm:
>
> - The Representation Collapse Problem: The Parallel Transformer formulates the block computation as $y = x + \mathrm{MHA}(\mathrm{LN}(x)) + \mathrm{MLP}(\mathrm{LN}(x))$. Crucially, this places it firmly within the PreNorm architectural family. Because the main residual path ($x$) bypasses normalization entirely, Parallel Transformers still suffer from unbounded variance growth ($\mathcal{O}(L)$) as depth increases. Consequently, they are entirely susceptible to the deep-layer representation collapse that SpanNorm mathematically prevents through its block-level bounding normalization.
>
> - Sequential Feature Refinement: By executing the MHA and MLP in parallel, Parallel Transformers decouple the two operations to gain a ~15% speedup. However, this means the MLP cannot condition on the attention output of that same block. SpanNorm preserves the serial refinement of standard Transformers (where the MLP processes the MHA output) while achieving the same stable gradient propagation through its spanning residual connection.
>
> > Comparison with ResiDual
>
> Thanks for highlighting that ResiDual also using the benefits of PreNorm and PostNorm to avoid gradient vanishing and representation collapse. However, our SpanNorm differs from two-folds:
>
> - Single-Stream vs. Dual-Stream Complexity: ResiDual prevents architectural pathologies by maintaining two concurrent residual streams (a PreNorm pathway and a PostNorm pathway) that run in parallel throughout the network. While effective, this dual-stream design fundamentally alters the Transformer topology and increases the memory footprint, as the network must track, compute, and fuse two distinct hidden states per layer.
>
> - Zero-Overhead Plug-and-Play: SpanNorm achieves the exact same theoretical guarantees—bounded signal variance and mitigated gradient decay via a standard single-stream topology. By simply extending the residual span and applying a single PostNorm-style computation at the block's exit, SpanNorm remains strictly zero-overhead. It requires no dual-state tracking, making it an easier plug-and-play replacement for standard architectures.
>
> To address the reviewer's suggestion within the limited rebuttal window, we conducted an empirical comparison of SpanNorm against both Parallel Transformer and Dual Residual architectures. We evaluated these models using the 5B configuration trained on 200B tokens. As shown in the table below, while the Parallel Transformer slightly outperforms the PreNorm baseline, it remains inferior to SpanNorm across most metrics. Dual Residual underperforms PreNorm in current setting, we suspect that Dual Residual requires extensive, scenario-specific hyperparameter tuning for LLMs, which we are continuing to investigate.
>
> | **Model**            | **Wiki. ppl ↓** | **LMB. ppl ↓** | **LMB. acc ↑** | **PIQA acc ↑** | **Hella. acc_norm ↑** | **SciQ acc ↑** | **ARC-c acc_norm ↑** | **Wino. acc ↑** | **Avg. acc ↑** |
> | -------------------- | --------------- | -------------- | -------------- | -------------- | --------------------- | -------------- | -------------------- | --------------- | -------------- |
> | PreNorm              | 12.5            | 6.9            | 61.4           | 73.7           | 64.4                  | 90.7           | 34.4                 | 60.1            | 64.1           |
> | SpanNorm             | 11.8            | 5.8            | 64.2           | 75.7           | 66.9                  | 92.0           | 36.3                 | 64.2            | 66.6           |
> | Mix-LN               | 12.3            | 6.9            | 61.3           | 75.6           | 65.1                  | 90.1           | 34.1                 | 61.4            | 64.6           |
> | HybridNorm           | 12.1            | 6.5            | 62.7           | 75.1           | 66.2                  | 91.4           | 35.4                 | 61.3            | 65.4           |
> | Parallel Transformer | 12.3            | 6.7            | 61.3           | 75.4           | 65.1                  | 90.4           | 35.1                 | 62.7            | 65.0           |
> | Dual Residual        | 12.8            | 7.6            | 59.5           | 74.1           | 63.2                  | 89.1           | 33.7                 | 61.3            | 63.4           |

---

> > ### Author Rebuttal · Reviewer_EaQV · 2026-04-03
> >
> > I want to thank the authors for thoroughly resolving my questions. I also appreciate running such an expensive exepriment (5B) within the rebuttal window. I am inclined to raise my score to 'Accept'.
> >
> > I would also like to thank reviewer KTfP for their good catch on the theoretical limitations. I think they should be better highlighted in the paper.

---

> > > ### Author Response · Authors · 2026-04-04
> > >
> > > We sincerely thank you for the feedback and for recognizing our efforts in conducting the 5B scale experiments. We are glad that the empirical comparisons and theoretical clarifications addressed your concerns. Regarding the theoretical limitations mentioned by Reviewer KTfP, we completely agree that these insights add necessary nuance to the paper. We are currently drafting a revision to highlight these points in a dedicated section, alongside the supplementary experiments and constructive discussions from all reviewers, to ensure a more balanced and comprehensive final manuscript. Thank you for your support and for helping us improve the final manuscript.

---

### Official Review · Reviewer_PNm8 · 2026-03-12

**Soundness:** 3
**Presentation:** 3
**Significance:** 3
**Originality:** 2
**Overall Recommendation:** 5
**Confidence:** 4

**Summary:**

The paper proposes a simple but novel transformer normalization technique called SpanNorm. It mitigates the issues appear in PreNorm and PostNorm transformer architectures. Extensive experiments including dense and MoE models show consistent improvements compared to other normalization techniques.

**Compliance With Llm Reviewing Policy:**

Affirmed.

**Final Justification:**

The rebuttal has addressed all of my concerns, hence I keep the original rating for acceptance.

**Key Questions For Authors:**

1. Have the authors try to test the application of SpanNorm in GQA and MLA?
2. For Assumption 3.1, it would be helpful to discuss the validity of this assumption. Will this hold in most of the LLM training?

**Limitations:**

The authors did not discuss the limitation of the work.

**Strengths And Weaknesses:**

Strength:

1. The paper is well-structured and easy to understand. Everything is explained in details.
2. The experiments are extensive. It spans MoE/dense architectures and various model sizes. Comparison with other baselines are also provided. The authors also provide deep scaling experiments so that SpanNorm can also be used to train extremely deep transformers.
3. The structure modification is minimal but the improvements are consistent. SpanNorm shows consistent improvements compared to all other baselines in all scenarios.

Overall, I believe this is a solid work.

Weaknesses:

1. The initialization method in Section 4 is not new as the authors also mention Megatron uses this initialization method. And this technique can date back to the GPT-2 paper.
2. Though authors mention that SpanNorm can apply to modern transformer variants like GQA and MLA, such experiments are not included. Including such experiments may significantly strengthen the work.

---

> ### Author Rebuttal · Authors · 2026-03-30
>
> We are thrilled by the reviewer's positive evaluation and sincerely thank you for the strong score. We deeply appreciate your recognition of SpanNorm's simple yet effective structural design, as well as the extensiveness of our evaluations across dense models, Mixture-of-Experts, and extreme-depth scaling scenarios. We address your specific constructive points below , and will add a dedicated Limitations section in the camera-ready version as suggested.
>
> > Clarification on the Initialization Method
>
> We completely agree with the reviewer that the $\mathcal{O}(1/\sqrt{L})$ initialization scaling is an established technique with roots in GPT-2 and widespread use in frameworks like Megatron-LM. We do not claim the initialization method itself as a novel contribution, and we explicitly note its origin in Megatron-LM in Section 4. Our primary contribution in Section 4 is providing the formal theoretical proof (Theorem 4.1) demonstrating why this specific scaling is a strict mathematical requirement for the newly proposed SpanNorm architecture. By proving that the FFN variance must scale by $\mathcal{O}(1/L)$ to prevent deep-layer variance explosion , we bridge the gap between this well-known empirical practice and SpanNorm's specific gradient dynamics. We will revise the introduction of Section 4 to make this distinction sharper and explicitly cite the GPT-2 paper alongside Megatron-LM to ensure comprehensive historical attribution.
>
>
> > Experiments with Modern Transformer Variants (MLA & GQA)
>
> We are grateful for this suggestion, as demonstrating compatibility with modern attention mechanisms is crucial for adoption.
>
> - MLA: We would like to respectfully direct the reviewer to Appendix B, where we detail the configurations for our MoE experiments (whose results are presented in Table 1). These experiments were conducted using the DeepSeek-V3 small-scale architecture , which explicitly employs the MLA mechanism. Therefore, SpanNorm's consistent improvements in our MoE evaluations already serve as empirical validation of its compatibility and effectiveness with MLA. We will move a brief mention of this architectural detail into the main text of Section 5 to ensure it is immediately apparent to readers.
>
> - GQA: We fully agree that demonstrating SpanNorm on dense models utilizing GQA will significantly strengthen the comprehensiveness of our work. We conducted an ablation study substituting MHA with GQA. We utilized the 740M parameter configuration (24 Q heads, 8 KV heads) trained on 30B tokens. The results, summarized below, confirm that the structural advantages of SpanNorm translate seamlessly to GQA-equipped networks, yielding lower perplexity and higher average accuracy on downstream benchmarks compared to the PreNorm baseline.
>
> | **Model**      | **Wiki. ppl ↓** | **LMB. ppl ↓** | **LMB. acc ↑** | **PIQA acc ↑** | **Hella. acc_norm ↑** | **SciQ acc ↑** | **ARC-c acc_norm ↑** | **Wino. acc ↑** | **Avg. acc ↑** |
> | -------------- | --------------- | -------------- | -------------- | -------------- | --------------------- | -------------- | -------------------- | --------------- | -------------- |
> | PreNorm (GQA)  | 21.5            | 22.1           | 39.3           | 67.5           | 42.0                  | 80.0           | 23.2                 | 52.5            | 50.8           |
> | SpanNorm (GQA) | 20.4            | 21.5           | 40.5           | 68.3           | 43.5                  | 82.1           | 24.5                 | 52.1            | 51.8           |
>
> > Validity of Assumption 3.1
>
> To validate this, we tracked the empirical statistics of the pre-normalized sums for Attention ($\sigma_{Attn}$ ) and FFN ($\sigma_{FFN}$ ) across our 64-layer 5B model. Here, $\sigma$ remains bounded and >1 across all depths. Furthermore, the ratio of $\sigma_{Attn}$ to $\sigma_{FFN}$ remains tightly centered around 1.08 (min 0.96, max 1.19). This confirms that variance neither explodes nor vanishes, and the sub-layers behave homogeneously, fully supporting our theoretical simplification. We will include this empirical analysis in the final manuscript.
>
> | Layer | 4 | 8 | 12 | 16 | 20 | 24 | 28 | 32 | 36 | 40 | 44 | 48 | 52 | 56 | 60 | 64 |
> |-------|------|------|------|------|------|------|------|------|------|------|------|------|------|------|------|------|
> |$\sigma_{Attn}$ | 1.3302 | 1.1277 | 1.1543 | 1.2008 | 1.1771 | 1.2450 | 1.2154 | 1.2526 | 1.1063 | 1.1148 | 1.1481 | 1.3363 | 1.3185 | 1.2252 | 1.1950 | 1.3201 |
> |$\sigma_{FFN}$ | 1.1487 | 1.1791 | 1.1496 | 1.1183 | 1.0968 | 1.1394 | 1.1197 | 1.1390 | 1.0963 | 1.0923 | 1.1061 | 1.1209 | 1.1163 | 1.1082 | 1.0549 | 1.2007 |
> | Ratio | 1.16 | 0.96 | 1.00 | 1.07 | 1.07 | 1.09 | 1.09 | 1.10 | 1.01 | 1.02 | 1.04 | 1.19 | 1.18 | 1.11 | 1.13 | 1.10 |

---

> > ### Author Rebuttal · Reviewer_PNm8 · 2026-04-03
> >
> > I would like to thank the authors for the rebuttal and additional experiments. I still believe this is a solid work and I will keep my current rating.

---

### Official Review · Reviewer_KTfP · 2026-03-13

**Soundness:** 3
**Presentation:** 3
**Significance:** 3
**Originality:** 3
**Overall Recommendation:** 4
**Confidence:** 4

**Summary:**

The paper introduces a new variant of normal	zing activations as they go through transformer layers which combines both stability benefits of PreNorm and performance gains of PostNorm. The authors theoretically derive the appropriate depth-dependent scaling to avoid vanishing gradients. They empirically compare their norm to popular ones in the literature.

**Compliance With Llm Reviewing Policy:**

Affirmed.

**Key Questions For Authors:**

- I am always a bit suspicious with tables of results where a new method is overwhelmingly better than the rest of the baselines, could the authors comment on that? For instance, have they found instances where the baselines performed better? I can see in Figure 14 that PreNorm does just as well (perhaps better) than SpanNorm, could they explain and give their intuition of it? Another point, could they elaborate on why they chose a postNorm ratio $\alpha=25$%?
- I find the authors’ analysis on width scaling quite interesting. I am aware of this work Nait Saada et al 2025, where the authors show that, at initialization only, the activations resulting form a single attention layer transformation with one head suffers from rank collapse (in terms of stable rank) as the width grows, regardless of PreNorm. Could the authors check what the result would be when switching PreNorm by SpanNorm and add the result of such experiment in their appendix C or D? This would require one forward pass only at initialization.
-  I find the ablations of figures 5, 6 and 7 interesting. Could the authors provide us with the same plots for the variants of Table 3? Do we observe that all the other variants have this non uniform distribution over layers for the gradients?

**Limitations:**

yes

**Strengths And Weaknesses:**

Strengths:
- The paper is very well written, polished, well structured and easy to follow.
- The empirical validation is strong with various ablations on how well the introduced norm transfers to width, depth, learning rates, etc
- The topic under study is of high practical relevance and it is tackled with a fair amount of both theory and empirical analyses.

Weaknesses:
- I find the theoretical section weaker than the empirical one. For instance, the authors call their claim a « Theorem » but they present a rather non rigorous version of it: many assumptions used in the proof are omitted to make their approximations (eg, FFN is considered to have a linear activation line 576, the residual branch is assumed to be uncorrelated to the identity path, line   593). It is obviously completely fine to make such assumptions for the sake of the proof but they should be clearly stated in the main body rather than deep somewhere in the proofs.
- Similarly, some theoretical derivations (eg 9 or 10) would benefit from further explanations. Could the authors please provide such explanations and add them to the main text where appropriate

---

> ### Author Rebuttal · Authors · 2026-03-30
>
> We sincerely thank the reviewer for recognizing the high practical relevance, strong empirical validation, and clear writing of our work. We deeply appreciate your constructive feedback regarding the theoretical presentation and the suggested additional analyses. We would like to address your concerns as below:
>
> ### Response to the weakness.
> > Strengthening the Theoretical Presentation & Assumptions
>
> We completely agree that the assumptions underlying Theorem 4.1 should be front and center. In the interest of space, we originally placed the assumptions regarding the identity activation function and the uncorrelated nature of the residual and identity paths in Appendix A. In the revised manuscript, we have moved these explicit assumptions directly into the main body in Section 4, immediately preceding the theorem, to ensure the mathematical rigor is entirely transparent.
>
> > Regarding Equations 9 and 10
>
> We will add expanded intuitive explanations in Section 3.3:
> - Equation 9: We will clarify that the $1/\sigma_l$ term drives the vanishing of the transformative branch. As the variance $\sigma_l$ grows with depth, this fraction shrinks, mathematically proving why deep PreNorm layers degenerate into near-identity functions and fail to learn distinct representations.
> - Equation 10: We will elaborate on how the sequential double-normalization in PostNorm creates a compounded scaling penalty. Because the gradient must pass through two normalization scalings ($\frac{1}{\sigma_{Y_i}}$ and $\frac{1}{\sigma_{X_i'}}$) per block, the attenuation compounds quadratically, explaining the catastrophic gradient vanishing observed empirically.
>
> ### Answer for the questions.
> > Resonse to Q1:
>
> - Performance Margins & Depth Scaling: We will clarify that SpanNorm is not universally superior (e.g., PreNorm wins Lambada/ARC-c at 740M). Crucially, SpanNorm's true advantage lies in deep scaling. By addressing PostNorm's instability and PreNorm's representation degradation, our method naturally shows its strongest improvements in deeper/larger regimes (5B, MoE, 128-layer).
> - Interpretation of Figure 14: Figure 14 does not claim SpanNorm's gradients are "better" than PreNorm's. It simply demonstrates that, unlike inherently stable PreNorm, SpanNorm requires our proposed Scale Init to avoid standard PostNorm's catastrophic gradient decay. Scale Init allows SpanNorm to match PreNorm's healthy gradient flow, enabling its superior downstream performance.
> - On the Mix-LN PostNorm ratio (25%): For the 64-layer 5B model,  we chose the 25% PostNorm ratio for two reasons. First, this ratio follows the setting used in the original Mix-LN paper, making the comparison faithful to the prior baseline. Second, we also tested it and found 12.5% PostNorm gave a higher LM loss than 25% (1.9878 v.s. 1.9794 ). And 50% became unstable and diverged.
>
> > Q2: Width Scaling and Rank Collapse (Nait Saada et al., 2025)
>
> We sincerely thank the reviewer for pointing us to Nait Saada et al. (2025). Following this suggestion, we ran the requested initialization-time stable-rank diagnostic and will include the results in the revised appendix.
> - Architectural Distinction: Nait Saada et al. developed their theory for bidirectional attention. In our autoregressive causal setting, the lower-triangular mask introduces deterministic zeros, breaking their i.i.d. Markov assumptions. We therefore use their analysis strictly as a diagnostic template, rather than a directly applicable theorem.
> - Single-Layer Diagnostic: As context width $T$ increases, PreNorm and SpanNorm exhibit remarkably similar stable-rank decay. This behavior is expected. As shown in Nait Saada et al.'s Figure 4a, modifying architectures outside the attention mechanism cannot fully resolve width-wise collapse without changing the Softmax kernel.
> - SpanNorm's Role: These results clarify our method's boundary. SpanNorm does not attempt to fix the static initialization gap driven by Softmax. Instead, it effectively resolves dynamic issues during actual training: the gradient vanishing of PostNorm and the representation collapse of PreNorm.
>
> > Q3: Additional Ablations for Table 3 Variants (Figures 5, 6, 7)
>
> This is a valuable addition to our empirical analysis. We are generating the representation similarity (Figs 5 & 6) and gradient norm distribution (Fig 7) plots for the 128-layer baseline variants from Table 3  (Mix-LN and HybridNorm).
>
> To directly answer your question: Yes, we do observe highly non-uniform gradient distributions for the failing variants. For instance, because HybridNorm completely diverged in the ultra-deep 128-layer regime, its gradient norms exhibit catastrophic decay in the early layers, mirroring the failure modes of standard PostNorm. These additional plots will be added to the appendix to comprehensively document the failure modes of the baselines at scale.

---

> > ### Author Rebuttal · Reviewer_KTfP · 2026-04-04
> >
> > Thank you for your engagement and for answering so thoroughly to all reviewers’ concerns and questions.
> > I find your analysis interesting and would keep my positive assessment of your work. One last thing, could you please turn your answer to my Q2 as a small paragraph to your appendix please? I do think this could resonate within the community.
> >
> > Thanks!

---

> > > ### Author Response · Authors · 2026-04-04
> > >
> > > We sincerely thank the reviewer for the continued engagement, the positive assessment of our work, and the constructive suggestion. We agree that the discussion on width scaling and rank collapse provides valuable context. As requested, we will gladly incorporate our response to Q2 into the appendix of the revised manuscript. This section will detail the architectural distinctions, the single-layer diagnostic, and SpanNorm’s specific role in addressing dynamic training issues versus initialization gaps. We will also cite this work as a valuable analytical resource. Additionally, all new experiments conducted during the rebuttal phase (all reviewers) will be included in the revision. Thank you again for helping us improve our work!

---

### Official Review · Reviewer_5hXG · 2026-03-18

**Soundness:** 2
**Presentation:** 2
**Significance:** 2
**Originality:** 2
**Overall Recommendation:** 3
**Confidence:** 4

**Summary:**

This paper proposes SpanNorm, a normalization method for Transformers, that aims to balance training stability and performance. By introducing a block-level residual connection with PostNorm-style normalization, SpanNorm combines the stability of PreNorm with the performance benefits of PostNorm. The authors provide theoretical analysis and empirical results showing improved stability and performance.

**Compliance With Llm Reviewing Policy:**

Affirmed.

**Key Questions For Authors:**

Q1. Could the authors provide a more systematic description of the hyperparameter tuning protocol for all compared normalization variants, especially learning rate, warmup, and any architecture-specific stabilization choices?

I am not yet convinced by the LR choices in the current comparison. Since even a 2x LR difference can cause a meaningful performance gap, it would be helpful if the authors could clarify how the LR for each baseline was selected and whether all methods were tuned under a comparable budget. In particular, I think the claim that the proposed method is stronger in performance would be more convincing if the baselines were also tested with appropriately tuned LRs, rather than only with a small set of selected values.

A clear response here would increase my confidence that the observed gains are due to the method itself rather than tuning differences.

Q2) How sensitive is SpanNorm relative to PreNorm, PostNorm, and other baselines to learning rate and optimization settings?

Q3) Could the authors clarify more precisely how SpanNorm should be positioned relative to prior hybrid normalization / residual-design approaches?

**Limitations:**

yes

**Strengths And Weaknesses:**

# Strengths
The paper gives a clear discussion of the pros and cons of the two dominant normalization designs, PreNorm and PostNorm, and makes a reasonable case for why the stronger side of PostNorm is worth pursuing. Based on that motivation, it proposes a method that aims to theoretically mitigate the main weakness of PostNorm. The paper also presents evidence in multiple forms. Main result tables show that the proposed method performs well, and the visual analyses, as commonly seen in normalization papers, help illustrate why the method may behave better than PreNorm or other normalization variants.

# Weaknesses
I am not fully convinced that the experimental study is broad enough. It is well known that the best learning rate and optimization setup can differ substantially depending on the normalization placement and residual architecture, but the paper does not seem to give enough evidence that this hyperparameter space was explored carefully for all baselines. This leaves open the possibility that the proposed method may have benefited from a more favorable training setup. I also think the paper shows mitigation of the vanishing-gradient issue in PostNorm rather than a full resolution, so I still have doubts about whether this architecture will reliably maintain both stability and performance as depth increases further. In addition, the method feels fairly close to prior work in spirit, the empirical gains are not especially large, and the paper does not yet make a strong case for why this should clearly be preferred over existing alternatives.

Overall, the idea is reasonable and the paper is fairly well executed, but the novelty and empirical, theoretical support do not feel strong enough to justify acceptance.

---

> ### Author Rebuttal · Authors · 2026-03-30
>
> We sincerely thank you for your constructive feedback. Your insightful questions regarding hyperparameter tuning fairness and our architectural positioning are crucial and have helped us significantly strengthen the paper. We have carefully addressed each of your concerns below.
>
> > Response to Q1 & Q2: Hyperparameter Tuning Protocol and LR Sensitivity
>
> 1. Tuning Protocol for Main Experiments (5B & 6.5B): To avoid disadvantaging the baselines, we did not force a shared LR when it was clearly suboptimal for them. SpanNorm and PreNorm used the same standard peak LRs (2e-4) and the same warmup setting, so SpanNorm does not benefit from a special LR choice relative to PreNorm. For HybridNorm and Peri-LN, we instead report their stronger stable settings at a smaller LR ($1 \times 10^{-4}$), because $2 \times 10^{-4}$ was empirically suboptimal for them under our setup. For instance, HybridNorm suffered from severe gradient vanishing at the $2 \times 10^{-4}$ LR, while for Peri-LN, we empirically found that a $1 \times 10^{-4}$ learning rate yielded significantly better convergence and better LM loss than $2 \times 10^{-4}$ (1.988 vs. 2.016).
>
> 2.  Matched-Budget LR Sweep (740M):
> To provide a comprehensive view of LR sensitivity, we conducted a new matched-budget LR sweep on the 740M model. The validation losses are summarized below:
>
> | Learning Rate (Log Scale) | SpanNorm         | PreNorm Baseline | PostNorm Baseline |
> | ----------------------------- | -------------------- | -------------------- | --------------------- |
> | 8.0e-5                        | 2.3130               | 2.3263               | 2.3487                |
> | 1.6e-4                        | 2.2454               | 2.2780               | **2.2705**   |
> | 3.2e-4                        | **2.2326**  | 2.2711               | Diverged              |
> | 6.4e-4                        | Diverged             | **2.2598**  | Diverged              |
> | 1.3e-3                        | -                    | 2.3350               | -                     |
>
> This sweep clearly demonstrates two key facts:
>
> - Superiority: SpanNorm consistently outperforms PreNorm across shared stable LRs, achieving the global optimum (2.2326).
>
> - Wider Window: SpanNorm exhibits significantly better stability than standard PostNorm, which diverges earlier (at 3.2e-4).
>
> 3. Extreme LR & Practicality:
> Because SpanNorm uses Post-style normalization on the aggregated output, it diverges at extreme LRs (6.4e-4) where PreNorm survives. However, as detailed in Appendix D.2 (Fig. 12), SpanNorm strictly adheres to width scaling principles. This allows reliable prediction of the maximum stable LR using a cheap proxy model, completely bypassing the need for expensive large-model sweeps in practical applications.
>
> > Response to Q3: Positioning Relative to Prior Hybrid Approaches
>
> We agree that the relationship to prior hybrid normalization / residual designs should be clarified more explicitly. Our intended claim is not that SpanNorm is the only hybrid architecture, but that it addresses the PreNorm/PostNorm trade-off from a different and more structural angle.
>
> A useful way to distinguish prior work from SpanNorm is by examining the specific design axis each modifies:
>
> - Prior Works (Axis of Placement and Augmentation): Existing methods largely operate by tweaking _where_ or _how many_ normalizations are applied. For instance, some methods mix PreNorm and PostNorm across different layers (e.g., Mix-LN) , while others insert additional normalization operations or localized hybrid rules within the block itself (e.g., HybridNorm, Peri-LN).
> - SpanNorm (Axis of Gradient-Transport Topology): In contrast, SpanNorm is driven by a different diagnosis. We identified that the main difficulty of PostNorm is not merely the placement of the norm, but that gradients are attenuated repeatedly as they serially traverse the two residual sublayers before reaching earlier representations . Therefore, SpanNorm does not simply "mix norms"; it fundamentally alters the gradient-transport topology by introducing a clean residual path that spans the entire block, preserving the PostNorm-style normalization strictly on the aggregated block output .
>
>  In this sense, SpanNorm is not just another interpolation between existing normalization placements, but a block-level reformulation of how residual paths and normalization interact. This difference is also reflected empirically. If the gain were merely due to another local hybrid tweak, one would expect improvements mainly at moderate scale/depth. Instead, SpanNorm remains strongest in the 5B comparison and, more importantly, continues to dominate in the 128-layer / 6.5B setting, where some prior hybrid variants already fail or lose their advantage. We view this as evidence that SpanNorm is addressing a more fundamental block-wise propagation issue, rather than providing only a cosmetic hybridization of existing normalization schemes.

---

> > ### Author Rebuttal · Reviewer_5hXG · 2026-04-06
> >
> > Thank you for the detailed rebuttal. It clarifies the intended tuning protocol and helps distinguish SpanNorm from prior hybrid normalization designs. However, my main concerns are only partially resolved. In particular, I still do not find the current evidence sufficient to rule out tuning-related advantages as a major factor in the reported gains. The new LR sweep is useful, but it still falls short of establishing a fair comparison against the most relevant competing baselines, especially HybridNorm and Peri-LN.
> >
> > My follow-up questions are:
> >
> > 1) Were similarly systematic matched-budget sensitivity checks performed for HybridNorm and Peri-LN, rather than only selecting their stronger stable learning rates?
> > 2) Given the current evidence, should the main takeaway be interpreted more cautiously as a robustness/stability advantage, rather than as a fully established best-case performance advantage over all closely related alternatives?

---

> > > ### Author Response · Authors · 2026-04-08
> > >
> > > **Response to Q1: Matched-Budget Sensitivity Checks and Tuning Protocol**
> > >
> > > Thank you for raising this issue. To address the concern about tuning-related advantage more directly, we clarify our protocol at the 5B scale and additionally provide a matched-budget LR sweep including HybridNorm and Peri-LN on the 740M proxy setting.
> > >
> > > 1. Tuning Protocol at the 5B Scale:
> > >
> > > For our 5B experiments comparing advanced LN variants (as presented in Table 2), we applied the standard default learning rate (2e-4) to both the PreNorm baseline and our proposed SpanNorm without any additional LR sweep. We did this intentionally to rigorously test SpanNorm's capability as a direct, drop-in replacement for PreNorm without requiring any modifications to the standard training recipe.
> > >
> > > For advanced baselines like HybridNorm and Peri-LN, we found that their specific architectural designs require architecture-specific LR tuning to reach their stronger settings. Consequently, we dedicated additional computational budget to systematically sweep learning rates for them (e.g., 1.5e-4 and 1e-4, as detailed in Appendix B).
> > >
> > > 2. Systematic Matched-Budget Sweep at the 740M Scale:
> > >
> > > To provide the direct, matched-budget sensitivity check you requested, which is computationally prohibitive to conduct as a full grid sweep at the 5B scale, we further ran the same LR sweep for HybridNorm and Peri-LN on the 740M setting.
> > >
> > > | **Learning Rate** | **SpanNorm** | **PreNorm** | **PostNorm** | **Peri-LN** | **HybridNorm** |
> > > | ----------------- | ------------ | ----------- | ------------ | ----------- | -------------- |
> > > | 8.0e-5            | 2.3130       | 2.3263      | 2.3487       | 2.2967      | 2.2830         |
> > > | 1.6e-4            | 2.2454       | 2.2780      | **2.2705**   | 2.2662      | **2.2488**     |
> > > | 3.2e-4            | **2.2326**   | 2.2711      | Diverged     | **2.2490**  | 2.2540         |
> > > | 6.4e-4            | Diverged     | **2.2598**  | Diverged     | 2.2660      | Diverged       |
> > > | 1.3e-3            | -            | 2.3350      | -            | 2.3327      | -              |
> > >
> > > This matched-budget sweep clarifies two key points:
> > >
> > > - SpanNorm still achieves the best global validation loss under the unified matched-budget sweep. Thus, its advantage is not explained simply by a favorable LR choice.
> > > - The sweep confirms that the optimal LR is architecture-dependent. In particular, for the closest advanced competitors (SpanNorm / HybridNorm / Peri-LN), the optima lie in a narrow $×2$ range between 1.6e-4 and 3.2e-4, whereas PreNorm prefers a larger LR. This supports our 5B tuning approach: allowing smaller LRs for Peri-LN and HybridNorm was necessary to reach their stronger settings, rather than an unfair disadvantage.
> > >
> > > We will add this comprehensive sweep table and clarify the budget allocation in the revised appendix.
> > >
> > > **Response to Q2: Interpretation of the Main Takeaway**
> > >
> > > We thank the reviewer for this nuanced perspective and agree that robustness and stability are critical advantages of SpanNorm. These benefits stem directly from a clear residual path, which is the primary strength of PreNorm architectures. This stability is most evident in our extreme depth-scaling stress tests, where SpanNorm scales smoothly from 32 to 512 layers.
> > >
> > > Furthermore, we wish to emphasize that this enhanced stability actively translates into a distinct performance advantage. By preserving a PostNorm-like computation flow, SpanNorm retains the strong representational capacity characteristic of those networks. Under systematic, matched-budget tuning  described above,  SpanNorm achieves the best observed peak validation performance among the compared variants. While we agree with the cautious approach of avoiding a universal "best-case" claim across all conceivable scenarios, we believe the current evidence supports that SpanNorm helps bridges this architectural gap, leveraging the training stability of PreNorm to unlock the superior performance typically exclusive to PostNorm.
> > >
> > > We hope these responses have fully addressed your concerns, and we kindly ask that you reconsider your evaluation of our work in light of these clarifications.

---

### Decision · Program_Chairs · 2026-04-30

**Decision:**

Accept (regular)

**Comment:**

SpanNorm offers a simple, zero-overhead normalization modification that spans the residual connection across the entire Transformer block, combining PreNorm's stability with PostNorm's representational strength. Three of four reviewers are satisfied post-rebuttal (4/5/5), and the holdout's tuning-confound concern (score 3) was substantively addressed by a matched-budget LR sweep where SpanNorm achieved the best validation loss without special tuning. The paper's strongest evidence, clean scaling to 128 layers where competitors diverge and monotonic gains to 512 layers, went uncontested. The theory relies on simplifying assumptions that limit its formal rigor, but the empirical contribution across dense and MoE settings at up to 5B scale is solid and practically useful. The camera-ready should foreground theoretical assumptions, add a limitations section, and incorporate rebuttal experiments.